# Training Fully Connected Neural Networks is $\exists\mathbb{R}$-Complete

**Daniel Bertschinger**
Department of Computer Science
ETH Zurich
Zurich, Switzerland
daniel.bertschinger@inf.ethz.ch

**Christoph Hertrich**[*]
Department of Mathematics
London School of Economics and Political Science
London, United Kingdom
c.hertrich@lse.ac.uk

**Paul Jungeblut**
Institute of Theoretical Informatics
Karlsruhe Institute of Technology
Karlsruhe, Germany
paul.jungeblut@kit.edu

**Tillmann Miltzow**
Department of Information and Computing Sciences
Utrecht University
Utrecht, The Netherlands
t.miltzow@uu.nl

**Simon Weber**
Department of Computer Science
ETH Zurich
Zurich, Switzerland
simon.weber@inf.ethz.ch

## Abstract

We consider the algorithmic problem of finding the optimal weights and biases for a two-layer fully connected neural network to fit a given set of data points, also known as *empirical risk minimization*. We show that the problem is $\exists\mathbb{R}$-complete. This complexity class can be defined as the set of algorithmic problems that are polynomial-time equivalent to finding real roots of a multivariate polynomial with integer coefficients. Furthermore, we show that arbitrary algebraic numbers are required as weights to be able to train some instances to optimality, even if all data points are rational. Our result already applies to fully connected instances with two inputs, two outputs, and one hidden layer of ReLU neurons. Thereby, we strengthen a result by Abrahamsen, Kleist and Miltzow [NeurIPS 2021]. A consequence of this is that a combinatorial search algorithm like the one by Arora, Basu, Mianjy and Mukherjee [ICLR 2018] is impossible for networks with more than one output dimension, unless NP $= \exists\mathbb{R}$.

## 1  Introduction

The usage of neural networks in modern computer science is ubiquitous. They are arguably the most powerful tool at our hands in machine learning [42]. One of the most fundamental algorithmic questions asks for the algorithmic complexity to actually train a neural network. For arbitrary network architectures, Abrahamsen, Kleist and Miltzow [3] showed that the problem is $\exists\mathbb{R}$-complete already for two-layer neural networks and linear activation functions. The complexity class $\exists\mathbb{R}$ is defined as the family of algorithmic problems that are polynomial-time equivalent to finding real roots of

---

[*]Moved to Université Libre de Bruxelles, Belgium, and Goethe-Universität Frankfurt, Germany, after submission of this article.

37th Conference on Neural Information Processing Systems (NeurIPS 2023).

multivariate polynomials with integer coefficients. Under the commonly believed assumption that $\exists\mathbb{R}$ is a strict superset of NP, this implies that training a neural network is harder than NP-complete problems.

The result by Abrahamsen, Kleist and Miltzow [3] has one major downside, namely that the network architecture is *adversarial*: The hardness inherently relies on choosing a network architecture that is particularly difficult to train. The instances by Abrahamsen, Kleist and Miltzow could be trivially trained with fully connected neural networks. This stems from the fact that they use the identity function as the activation function which reduces the problem to matrix factorization. While intricate network architectures, e.g. convolutional and residual neural networks, pooling, autoencoders and generative adversarial neural networks are common in practice, they are usually designed in a way that facilitates training rather than making it difficult [42]. We strengthen the result in [3] by showing $\exists\mathbb{R}$-completeness for *fully connected* two-layer neural networks. This shows that $\exists\mathbb{R}$-hardness does not stem from one specifically chosen worst-case architecture but is inherent in the neural network training problem itself. Although in practice a host of different architectures are used, fully connected two-layer neural networks are arguably the most basic ones and they are often part of more complicated network architectures [42]. We show hardness even for the case of fully connected two-layer ReLU neural networks with exactly two input and output dimensions.

Remarkably, with only one instead of two output dimensions, the problem is in NP. This follows from a combinatorial search algorithm by Arora, Basu, Mianjy and Mukherjee [6]. Our result explains why their algorithm was never successfully generalized to more complex network architectures: adding only a second output neuron significantly increases the computational complexity of the problem, from being contained in NP to being complete for $\exists\mathbb{R}$.

To achieve our result, our reduction follows a completely novel approach compared to the reduction in [3]. Instead of encoding polynomial inequalities into an adversarial network architecture, we make use of the underlying geometry of the functions computed by two-layer neural networks and utilize the fact that their different output dimensions have nonlinear dependencies.

**Outline.** We start by formally introducing neural networks, the training problem, and the existential theory of the reals in Section 2. Thereafter, we present our main results in Section 3. In Section 4 we provide different perspectives on how to interpret our findings, including an in-depth discussion of the strengths and limitations. We cover further related work in Section 5. Finally, we finish the main part of the paper by presenting the key ideas we use to prove our result in Section 6. The complete proof details are contained in the supplementary material.

## 2   Preliminaries

In this section we discuss the necessary definitions and preliminaries related to neural network training and the complexity class $\exists\mathbb{R}$.

**Neural Networks and Training.** All neural networks considered in this paper have a very simple architecture. For ease of presentation, we therefore do not define neural networks and their training problem in full generality here. Instead we restrict ourselves to the simple architectures we need.

**Definition 1** (Fully Connected Two-Layer Neural Network). A *fully connected two-layer neural network* $N = (S \cup H \cup T, E)$ is a directed acyclic graph (the *architecture*) with real-valued edge weights. The vertices, called *neurons*, are partitioned into the *inputs* $S$, the *hidden neurons* $H$ and the *outputs* $T$. All possible edges from $S$ to $H$ as well as all possible edges from $H$ to $T$ are present. Additionally, each hidden neuron has a real-valued *bias* and an *activation function* ($\mathbb{R} \to \mathbb{R}$).

The probably most commonly used activation function [6, 37, 42] is the *rectified linear unit (ReLU)* defined as $\mathrm{ReLU} : \mathbb{R} \to \mathbb{R}, x \mapsto \max\{0, x\}$. See Figure 1 for a small fully connected two-layer ReLU neural network.

Given a neural network architecture $N$ as defined above, assume that we fixed an ordering on $S$ and $T$. Then $N$ realizes a function $f(\cdot, \Theta) : \mathbb{R}^{|S|} \to \mathbb{R}^{|T|}$, where $\Theta$ denotes the weights and biases that parameterize the function. For $x \in \mathbb{R}^{|S|}$ we define $f(\cdot, \Theta)$ inductively: The $i$-th input neuron forwards the $i$-th component of $x$ to all its outgoing neighbors. Each hidden neuron forms the

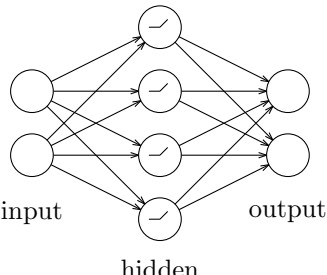

Figure 1: A fully connected two-layer neural network as studied in this paper. The symbol inside the hidden neurons expresses the $\mathrm{ReLU}$ activation function.

weighted sum over all incoming values, adds its bias, applies the activation function to this sum and forwards it to all outgoing neighbors. An output neuron again forms the weighted sum over all incoming values but does not add a bias and does not apply any activation function.

**Definition 2** (TRAIN-F2NN)**.** Training a fully connected two-layer neural network is the following decision problem, denoted by TRAIN-F2NN:

**Input:** A 5-tuple $(N, \varphi, D, \gamma, c)$, where

- $N = (S \cup H \cup T, E)$ is the network architecture,
- $\varphi : \mathbb{R} \to \mathbb{R}$ is an activation function to be used at all neurons in the hidden layer,
- $D$ is a set of $n$ data points of the form $(x; y) \in \mathbb{Q}^{|S|} \times \mathbb{Q}^{|T|}$,
- $\gamma \in \mathbb{Q}_{\geq 0}$ is the target error, and
- $c : \mathbb{R}^{|T|} \times \mathbb{R}^{|T|} \to \mathbb{R}_{\geq 0}$ is a loss function.

**Question:** Are there weights and biases $\Theta$ such that $\sum_{(x;y) \in D} c(f(x, \Theta), y) \leq \gamma$?

The data points $D$ and the target error $\gamma$ are part of the input, which has finite length (in some chosen encoding). We restrict them to be rational numbers because these are straightforward to encode by noting their numerator and denominator in binary, in contrast to much more complicated encodings required for arbitrary (algebraic) real numbers.

We further require that the loss function is *honest*, meaning that it returns zero if and only if the data is fit exactly. We will see that this is the only requirement on the loss function to prove $\exists \mathbb{R}$-hardness of the zero-error case ($\gamma = 0$).

**Existential Theory of the Reals.** The complexity class $\exists \mathbb{R}$ (pronounced as "ER") has gained a lot of interest in recent years. It is defined via its canonical complete problem ETR (short for *Existential Theory of the Reals*) and contains all problems that polynomial-time many-one reduce to it. In an ETR instance we are given an integer $n$ and a sentence of the form

$$\exists X_1, \ldots, X_n \in \mathbb{R} : \varphi(X_1, \ldots, X_n),$$

where $\varphi$ is a well-formed and quantifier-free formula consisting of polynomial equations and inequalities in the variables with integer coefficients encoded in binary, and the logical connectives $\{\wedge, \vee, \neg\}$. The goal is to decide whether this sentence is true. As an example consider the formula $\varphi(X, Y) :\equiv X^2 + Y^2 \leq 1 \wedge Y^2 \geq 2X^2 - 1$; among (infinitely many) other solutions, $\varphi(0, 0)$ evaluates to true, witnessing that this is a yes-instance of ETR. It is known that

$$\mathsf{NP} \subseteq \exists \mathbb{R} \subseteq \mathsf{PSPACE}$$

and it is widely believed that both inclusions are strict. Here the first inclusion follows because a SAT instance can easily be expressed as an equivalent ETR instance [83]. The second inclusion is highly non-trivial and follows from a result by Canny [17].

Note that the complexity of problems involving real numbers was studied in various contexts. To avoid confusion, let us emphasize that the underlying machine model for $\exists \mathbb{R}$ (over which sentences need to be decided and where reductions are performed in) is the standard binary word RAM (or equivalently, a Turing machine), just like for P, NP, and PSPACE. It is *not* the real RAM [33] or the Blum-Shub-Smale model [13].

# 3 Results

We are now ready to provide the formal statements of our results. The main result is as follows:

**Theorem 3.** *The problem* TRAIN-F2NN *is* ∃ℝ-*complete, even if:*

- *There are only two input neurons.*
- *There are only two output neurons.*
- *The number of data points is linear in the number of hidden neurons.*
- *The data has only* 13 *different labels.*
- *The target error is* $\gamma = 0$.
- *The* ReLU *activation function is used.*

Let us note that all of the restrictions in Theorem 3 together describe a special case of the general training problem (Definition 2). Proving that this special case is ∃ℝ-hard also implies ∃ℝ-hardness for the general problem, for example with more input/output neurons, more data points, more labels, arbitrary $\gamma \geq 0$, and possibly different activation functions.

Apart from settling the precise computational complexity status of the problem to train fully-connected neural networks, our result implies that any approach to solve TRAIN-F2NN to global optimality must involve more sophisticated methods than those used for typical NP-complete problems.

Additionally, our reduction to prove ∃ℝ-hardness also implies *algebraic universality*:

**Theorem 4.** *Let* $\alpha \in \mathbb{R}$ *be an algebraic number. Then there exists an instance of* TRAIN-F2NN, *which has a solution if the weights and biases* $\Theta$ *are restricted to* $\mathbb{Q}[\alpha]$, *but no solution when the weights and biases* $\Theta$ *are restricted to a field* $\mathbb{F}$ *not containing* $\alpha$.

Here, $\mathbb{Q}[\alpha]$ is the smallest field extension of $\mathbb{Q}$ containing $\alpha$. This means that for some training instances all global optima require irrational weights even if all data points are integral.

Algebraic universality is known to hold for various ∃ℝ-complete problems [4], however this is not an automatism: Algebraic universality cannot occur in problems where the solution space is open, which for example is the case in some problems of recognizing certain intersection graphs [18, 57].

# 4 Discussion

In this section, we discuss our results from various perspectives, pointing out strengths and limitations.

**Implications of ∃ℝ-Completeness.** It is widely believed that NP ⊊ ∃ℝ. From a theoretical standpoint, proving ∃ℝ-completeness of an NP-hard problem is therefore a valuable contribution because we are interested in the exact complexity of important algorithmic problems. Although this concerns only the worst case complexity, proving ∃ℝ-completeness also hints at the difficulties that need to be overcome when designing algorithms for the considered problem. For example, the continuous solution space of ∃ℝ-complete problems presents an algorithmic challenge. This is in contrast to problems in NP with a discrete or discretizable solution space, which can often be solved well by extremely optimized off-the-shelf tools like SAT- or MIP-solvers. No such general purpose tools are available for ∃ℝ-complete problems.

However, let us stress that ∃ℝ-completeness does not rule out hope for good heuristics: The ∃ℝ-complete art gallery problem can be solved well in practice via custom heuristics, some of these also in combination with integer programming solvers [24]. Under additional assumptions, performance guarantees can be proven [48]. However, these heuristics are specifically tailored towards the art gallery problem. Identifying reasonable assumptions for other ∃ℝ-complete problems in order to obtain good heuristics (possibly even with performance guarantees) is an important open question.

One meta-heuristic that can often be used to get "good" solutions for ∃ℝ-complete problems is gradient descent. Probably the most prominent example in this context is training neural networks, where a bunch of different gradient descent variants powered by backpropagation are nowadays capable of training neural networks containing millions of neurons. In general, we do not get any approximation or runtime guarantees when using gradient descent, but under the right additional assumptions proving such guarantees is sometimes possible [15]. Gradient descent has also been applied to ∃ℝ-complete problems from other areas, for example to graph drawing [5]. Still though,

these gradient descent approaches are tailored towards the specific problem at hand. It would be very desirable to have general purpose solvers, similar to SAT- or MIP-solvers for problems in NP.

**Relation to Learning Theory.** In this paper we purely focus on the computational complexity of empirical risk minimization, that is, minimizing the *training error*. In the machine learning practice, one usually desires to achieve low *generalization error*, which means to use the training data to achieve good predictions on unseen test samples.

To formalize the concept of the generalization error, one needs to combine the computational aspect with a statistical one. There are various models to do so in the literature, the most famous one being *probably approximately correct* (PAC) learnability [80, 87]. While empirical risk minimization and learnability are two different questions, they are strongly intertwined; see Section 5 for related work in this context. Despite the close connections between empirical risk minimization and learnability, to the best of our knowledge, the $\exists\mathbb{R}$-hardness of the former has no direct implications on the complexity of the latter. Still, since empirical risk minimization is the most common learning paradigm in practice, our work is arguably also interesting in the context of learning theory.

**Required Precision of Computation.** An implication of $\exists\mathbb{R}$-hardness of TRAIN-F2NN is that for some instances every set of weights and biases exactly fitting the data needs large precision, actually a superpolynomial number of bits to be written down. The algebraic universality of TRAIN-F2NN (Theorem 4) strengthens this by showing that exact solutions require algebraic numbers. This restricts the techniques one could use to obtain optimal weights and biases even further, as it rules out numerical approaches (even using arbitrary-precision arithmetic), and shows that symbolic computation is required. In practice, we are often willing to accept small additive errors when computing $f(\cdot, \Theta)$ and therefore also do not require $\Theta$ to be of such high precision. In other words, rounding the weights and biases $\Theta$ to the first "few" digits after the comma may be sufficient. This might allow to place the problem of *approximate* neural network training in NP. Yet, we are not aware of such a proof and we consider it an interesting open question to establish this fact thoroughly. Let us note that a similar phenomenon appears with many other $\exists\mathbb{R}$-complete problems [33]: While an exact solution $x$ requires high precision, there is an approximate solution $\tilde{x}$ close to $x$ that needs only polynomial precision. However, guessing the digits of the solution in binary is in no way a practical algorithm to solve these problems. Moreover, historically, $\exists\mathbb{R}$-completeness seems to be a strong predictor that finding these approximate solutions is difficult in practice [25, 33]. Therefore, we consider $\exists\mathbb{R}$-completeness to be a strong indicator of the theoretical and practical algorithmic difficulties to train neural networks.

Related to this, Bienstock, Muñoz, and Pokutta [9] use the above idea to discretize the weights and biases to show that, in principle, arbitrary neural network architectures can be trained to approximate optimality via linear programs with size linear in the size of the data set, but exponential in the architecture size. While being an important insight, let us emphasize that this does not imply NP-membership of an approximate version of neural network training.

**Number of Input Neurons.** In practice, neural networks are often trained on high dimensional data, thus having only two input neurons is even more restrictive than the practical setting. Note that we easily obtain hardness for higher input dimensions by simply placing all data points of our reduction into a two-dimensional subspace. The precise complexity of training fully connected two-layer neural networks with only one-dimensional input and multi-dimensional output remains unknown. While this setting does not have practical relevance, we are still curious about this open question from a purely mathematical perspective.

**Number of Output Neurons.** As discussed earlier, if there is only one output neuron instead of two, then the problem is known to be contained in NP [6]. Our reduction can easily be extended to the case with more than two output neurons by padding all output vectors with zeros. Thus, the complexity classification is complete with respect to the number of output neurons.

**Number of Hidden Neurons.** Consider the situation that the number $m$ of hidden neurons is larger than the number $n$ of data points. If there are no two contradicting data points $(x_1, y_1)$ and $(x_2, y_2)$ with $x_1 = x_2$ but $y_1 \neq y_2$, then we can always fit all the data points exactly [91]. Thus, we need at least a linear number of data points in terms of $m$ to be able to get $\exists\mathbb{R}$-completeness. While we already achieve a linear number, we wonder whether our result could be strengthened further by

showing $\exists\mathbb{R}$-completeness with $n \le (1 + \varepsilon)m$, for every fixed $\varepsilon$. Note that by adding additional data points, the ratio between $n$ and $m$ can be made arbitrarily large. Thus our reduction holds also for all settings in which $m$ is (asymptotically) much smaller than $n$.

**Number of Output Labels.** The number of labels, that is, different values for $y \in \mathbb{Q}^{|T|}$ in the data set $D$, used in our reduction is just 13. A small set of possible labels shows relevance of our result to practice, where we are often confronted with a large number of data points but a much smaller number of labels, for instance, in classification tasks. The set of labels used by us is

$$\big\{(-2,6), (-1,-1), (-1,0), (0,-1), (0,0), (0,3), (2,2), (3,3), (3,6), (4,4), (6,-2), (6,6), (10,10)\big\}.$$

If all labels are contained in a one-dimensional affine subspace the problem is in NP, as then they can be projected down to one-dimensional labels and the problem can be solved with the algorithm by Arora, Basu, Mianjy and Mukherjee [6]. As any two labels span a one-dimensional affine subspace, the problem can only be $\exists\mathbb{R}$-hard for at least three different affinely independent output labels.

We think it is not particularly interesting to close the gap between 13 and 3 output labels, but it would be interesting to investigate the complexity of the problem when output labels have more structure. For example, in classification tasks one often uses *one-hot encodings*, where for $k$ classes, a $k$-dimensional output space is used and all labels have the form $(0, \dots, 0, 1, 0, \dots, 0)$. Note that in this case, at least three output dimensions are needed to obtain three different labels.

**Target Error.** For simplicity, we show hardness for the case with target error $\gamma = 0$. However, in practice it is generally not required to fit the data exactly. It is not too difficult to see that we can easily modify the value of $\gamma$ by adding inconsistent data points that can only be fit best in exactly one way. The precise choice of these inconsistent data points heavily depends on the loss function. In conclusion, for different values of $\gamma$, the decision problem does not get easier.

**Activation Functions.** Let us point out that the ReLU activation function is currently still the most commonly used activation function in practice [6, 37, 42]. Our methods are probably easily adaptable to other piecewise linear activation functions, such as leaky ReLUs. Having said that, our methods are *not* applicable to other types of activation functions, such as Sigmoid, soft ReLU or step functions. We want to point out that TRAIN-F2NN (and even training of arbitrary other architectures) is in NP in case a step function is used as the activation function [56]. For the Sigmoid and soft ReLU function it is not even clear whether the problem is decidable, as trigonometric functions and exponential functions are not computable even on the real RAM [33, 70].

**Lipschitz Continuity and Regularization.** The set of data points created in the proof of Theorem 3 is intuitively very tame. We formalize this by proving that for yes-instances there exists a $\Theta$ such $f(\cdot, \Theta)$ is Lipschitz continuous for a small Lipschitz constant $L$ (see Remark 23 in the supplementary material for details). This is also related to overfitting and regularization [43], the purpose of the latter being to prefer simpler functions over more complicated ones. Being Lipschitz continuous with a small Lipschitz constant essentially means that the function is pretty flat. It is particularly remarkable that we can show hardness even for small Lipschitz constants since Lipschitz continuity has been a crucial assumption in several recent results about training *and* learning ReLU networks, for example in [9, 20, 38].

**Other Architectures.** While we consider fully connected two-layer networks as the most important case, we are also interested in $\exists\mathbb{R}$-hardness results for other network architectures. Specifically, fully connected three-layer neural networks and convolutional neural networks are interesting. While it is hard to imagine that more complicated architectures are easier to train, a formal proof of this intuition would strengthen our result and show that $\exists\mathbb{R}$-completeness is a robust phenomenon, in other words independent of a choice of a specific network type.

## 5 Further Related Work

**Complexity of Training Neural Networks.** It is well-known that minimizing the training error of a neural network is a computationally hard problem for a large variety of activation functions and architectures [80]. For ReLU networks, NP-hardness, parameterized hardness and inapproximability

results have been established even for the simplest possible architecture consisting of only a single ReLU neuron [14, 27, 35, 40]. While all these results require non-constant input-dimension, Froese and Hertrich [34] showed that it is also NP-hard to train a two-layer ReLU network with input dimension two and output dimension one. On the positive side, the seminal algorithm by Arora, Basu, Mianjy and Mukherjee [6] solves empirical risk minimization for two-layer ReLU networks and one-dimensional output to global optimality, placing the problem in NP. It was later extended to a more general class of loss functions by Froese, Hertrich and Niedermeier [35]. The running time is exponential in the number of neurons in the hidden layer and in the input dimension, but polynomial in the number of data points if the former two parameters are considered to be a constant. This NP-containment of TRAIN-F2NN with one-dimensional output is in sharp contrast to our $\exists\mathbb{R}$-completeness result for two-dimensional outputs.

While minimizing training and generalization errors are different problems, the hardness of the former also imposes challenges on the latter. Strategies to circumvent hardness from the perspective of learning theory include allowing improper learners, restricting the type of weights allowed in a neural network, or imposing assumptions on the underlying distribution. For example, Chen, Klivans and Meka [20] show fixed-parameter tractability of learning a ReLU network under some assumptions including Gaussian data and Lipschitz continuity of the network. We refer to [7, 19, 28, 38, 41, 39] as a non-exhaustive list of other results about (non-)learnability of ReLU networks in different settings.

**Expressivity of ReLU Networks.** It is essential for our reduction to understand the classes of functions representable by certain ReLU network architectures. So-called *universal approximation theorems* state that a single hidden layer (with arbitrary width) is already sufficient to approximate every continuous function on a bounded domain with arbitrary precision [22, 52]. However, deeper networks require much fewer neurons to reach the same expressive power, yielding a potential theoretical explanation of the dominance of deep networks in practice [6, 31, 45, 47, 58, 67, 69, 72, 85, 86, 89]. Other related work includes counting and bounding the number of linear regions [46, 64, 65, 68, 69, 79], classifying the set of functions *exactly* representable by different architectures [6, 26, 44, 49, 50, 51, 66, 92], or analyzing the memorization capacity of ReLU networks [88, 90, 91]. Huchette, Muñoz, Serra, and Tsay [53] provide a survey on the interactions of neural networks and polyhedral geometry, including implications on training, verification, and expressivity.

**Existential Theory of the Reals.** The complexity class $\exists\mathbb{R}$ gains its importance by numerous important algorithmic problems that have been shown to be complete for this class in recent years. The name $\exists\mathbb{R}$ was introduced by Schaefer in [73] who also pointed out that several NP-hardness reductions from the literature actually implied $\exists\mathbb{R}$-hardness. For this reason, several important $\exists\mathbb{R}$-completeness results were obtained before the need for a dedicated complexity class became apparent.

Common features of $\exists\mathbb{R}$-complete problems are their continuous solution space and the nonlinear relations between their variables. Important $\exists\mathbb{R}$-completeness results include the realizability of abstract order types [63, 83] and geometric linkages [74], as well as the recognition of geometric segment [57, 60], unit-disk [55, 61], and ray intersection graphs [18]. More results appeared in the graph drawing community [30, 32, 59, 75], regarding polytopes [29, 71], the study of Nash-equilibria [8, 10, 11, 36, 76], matrix factorization [21, 78, 81, 82], or continuous constraint satisfaction problems [62]. In computational geometry, we would like to mention the art gallery problem [2, 84] and covering polygons with convex polygons [1].

Recently, the community started to pay more attention to higher levels of the *real polynomial hierarchy*, which also capture several interesting algorithmic problems [12, 16, 23, 30, 54, 77].

## 6  Proof Ideas

To show $\exists\mathbb{R}$-hardness, we reduce from a classical version of ETR called ETR-INV. An instance of ETR-INV consists of real variables and a conjunction of constraints in these variables, where each constraint is either and addition constraint $X + Y = Z$ or an inversion constraint $X \cdot Y = 1$. Given such an instance, we construct a TRAIN-F2NN instance that models the variables and whose training corresponds to satisfying all addition and inversion constraints.

**Variables.**    A natural candidate for encoding variables are the weights and biases of the neural network. However, those did not prove to be suitable for our purposes. The main problem with using the parameters of the neural network as variables is that the same function can be computed by many neural networks with different combinations of these parameters. We are not aware of an easy way to normalize the parameters.

To circumvent this issue, we work with the functions representable by fully connected two-layer neural networks directly. We frequently make use of the geometry of these functions. For now, it is only important to understand that each hidden ReLU neuron encodes a continuous piecewise linear function with exactly two pieces (both of constant gradient). These two pieces are separated by a so-called *breakline*. Now, if we have $m$ middle neurons, their individually encoded functions add up such that the function computed by the whole neural network is a continuous piecewise linear function with at most $m$ breaklines. Between these breaklines all pieces of the function have constant gradient.

To keep things simple for now, let us first consider a neural network with only one input and one output neuron. We place a series of data points $(x_i, y_i) \in \mathbb{R}^2$ as seen in Figure 2. All continuous piecewise linear functions $f(\cdot, \Theta)$ computed by a neural network with only four hidden neurons (i.e., only four breaklines and therefore at most five pieces) that fit these data points exactly must be very similar. In fact, they can only differ in one degree of freedom, namely the slope of the piece going through point $p$. In our construction, this slope represents the value of a variable. The whole set of data points enforcing this configuration is called a *variable gadget*.

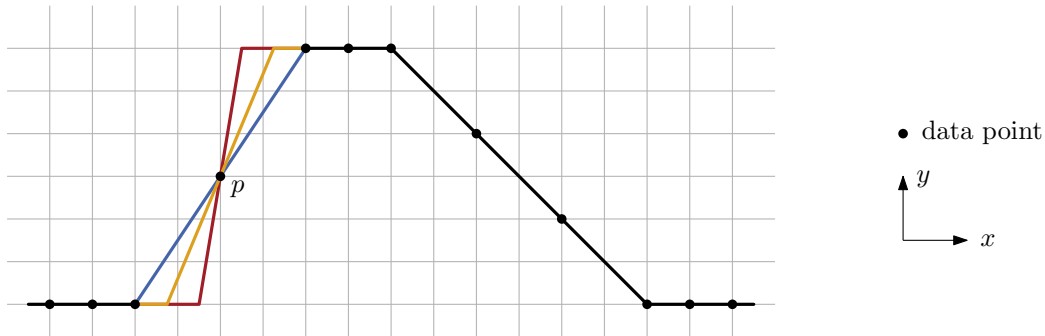

Figure 2: The value of $f(\cdot, \Theta)$ is fixed (black part), except for the segment through data point $p$. The red, orange and blue segments are just three out of uncountably many possibilities. Its slope can be used to encode a real-valued variable.

**Linear Dependencies.**    The key insight for encoding constraints between variables is that we can relate the values of several variable gadgets by a data point: By placing a data point $p$ at a location where several variable gadgets overlap, each of the variable gadgets contributes its part towards fitting $p$. The exact contribution of each variable gadget depends on its slope. Consequently, if one variable gadget contributes more, the others have to proportionally contribute less. This enforces linear dependencies between different variable gadgets and can be used to design addition and copy gadgets.

To be able to intersect multiple variable gadgets, we need a second input dimension. Each variable gadget as seen in Figure 2 is now extended into a stripe in $\mathbb{R}^2$, with Figure 2 showing only an orthogonal cross section of this stripe. See Figure 3 for two intersecting variable gadgets. Much of the technical difficulties lie in the subtleties to enforce the presence of multiple (possibly intersecting) gadgets using a finite number of data points.

To construct an addition gadget we need to be able to encode linear relations between three variables. In $\mathbb{R}^2$, three lines (or stripes) usually do not intersect in a single point. Thus, for addition we have to carefully place them to guarantee such a single intersection point. This might involve creating copies of a variable gadget, which is a linear relation between just two variable gadgets.

**Inversion Constraint.**    Within only a single output dimension, we are not able to encode nonlinear constraints [6]. We therefore add a second output dimension, which implies that the neural network

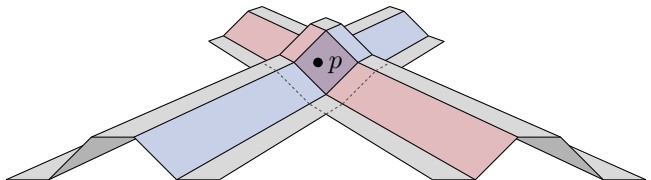

Figure 3: Two variable gadgets (slopes of the blue and red regions encode the values). Point $p$ lies in the intersection of both and can encode a linear relationship between them.

represents two functions $f^1(\cdot, \Theta)$ and $f^2(\cdot, \Theta)$. Consequently, we are allowed to use data points with two different output labels, one for each output dimension.

One important observation is that the locations of breaklines of $f = f(\cdot, \Theta) = (f_1(\cdot, \Theta), f_2(\cdot, \Theta))$ only depend on the weights and biases in the first layer of the neural network. Thus, every breakline is present in both functions $f^1$ and $f^2$, except if the weight to the corresponding output neuron is zero. This connects the two output dimensions in a way that we can encode nonlinear constraints.

We define an inversion gadget (realizing the constraint $X \cdot Y = 1$), which also corresponds to a stripe in $\mathbb{R}^2$. For simplicity, we only show a cross section here, see Figure 4. In each output dimension individually, the inversion gadget looks exactly like a variable gadget. The inversion gadget can therefore be understood as a variable gadget that carries two values.

We prove that by allowing only five breaklines in total, a function $f$ can only fit all data points exactly if $f^1$ and $f^2$ share three of their four breaklines (while both having one "exclusive" breakline each). This enforces a nonlinear dependency between the slopes of $f^1$ and $f^2$. By choosing the right parameters, this nonlinear relation models exactly an inversion constraint.

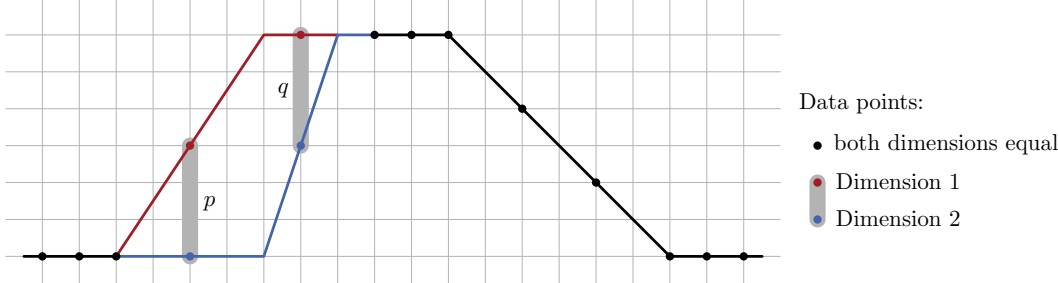

Figure 4: Data points $p$ and $q$ have different labels in the two output dimensions, enforcing that the slopes of the red and the blue pieces are related via a nonlinear dependency.

**Reduction.** Let us illustrate the reduction by giving a simple example. Note that this is not yet the "complete picture", which follows in the supplementary material. We start with the ETR-INV instance that is to decide whether the following sentence

$$\exists X_1, X_2, X_3, X_4 \in \mathbb{R} : (X_1 + X_2 = X_3) \wedge (X_1 + X_3 = X_4) \wedge (X_1 \cdot X_4 = 1) \wedge (X_4 \cdot X_3 = 1)$$

is true. This instance has four variables $X_1, X_2, X_3, X_4$ and four constraints: two additions and two inversions. Recall that every gadget corresponds to a stripe in the input space $\mathbb{R}^2$. See Figure 5 for the following construction (for better readability the stripes are drawn as lines).

- For each variable we add a variable gadget. We place all of these such that their corresponding stripes are parallel and do not overlap, see the horizontal lines in Figure 5.
- For each addition constraint we introduce three new variable gadgets, one per involved variable. We place these such that they have a common intersection point while also intersecting their corresponding variable gadget. In Figure 5, see the two bundles to the left. A data point at the triple intersection enforces the addition constraint, while data points labelled $\bullet_=$ encode that the values of the two intersecting variable gadgets are equal.
- Lastly, we add an inversion gadget for each inversion constraint and place it such that it intersects the variable gadgets of the two involved variables. See the two dashed lines in

Figure 5. Data points labelled $\bullet^{=_1}$ ($\bullet^{=_2}$) enforce that the values stored in the first (second) output dimension are equal.

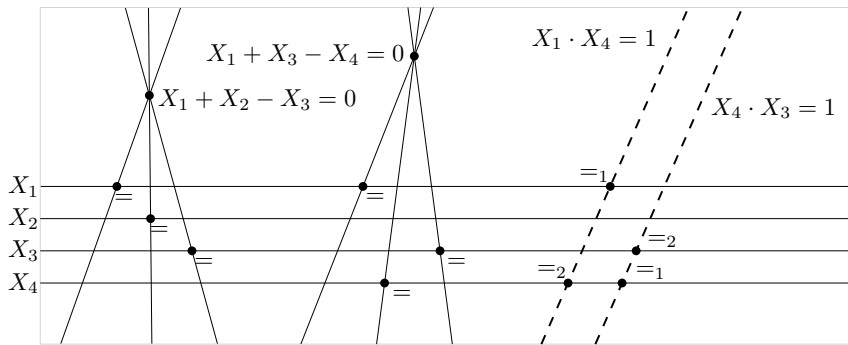

Figure 5: Overview of the global arrangement of the gadgets.

To see that above reduction is correct, assume that the sentence in the ETR-INV instance is indeed true. Then there exists real values for the four variables and these values can be used as the slopes of their corresponding variable gadgets. By the correctness of the individual gadgets (which is to be proven in the supplementary material) it follows that each data point is fit exactly.

Conversely, if all data points are fit exactly, then the correctness of the gadgets implies that the slopes of the variable gadgets give a solution to the ETR-INV instance. Intuitively, this holds because the addition and inversion constraints are encoded exactly by the gadgets of our construction.

**Outline of the Supplementary Material.** We sketch $\exists\mathbb{R}$-membership of TRAIN-F2NN in Appendix A before turning to $\exists\mathbb{R}$-hardness in Appendix B. As Appendix B is by far the longest part of the paper, it is further broken down into several parts: After a short reduction overview, we describe suitable abstractions to make defining and analyzing the gadgets easier. Then we introduce the gadgets formally one by one and in isolation before finally putting everything together. We finish the section by proving correctness of our reduction. Finally, we prove algebraic universality in Appendix C.

## Acknowledgments and Disclosure of Funding

Christoph Hertrich is supported by the European Research Council (ERC) under the European Union's Horizon 2020 research and innovation programme (grant agreements ScaleOpt–757481 and ForEFront–615640). Tillmann Miltzow is generously supported by the Netherlands Organisation for Scientific Research (NWO) under project numbers 016.Veni.192.250 and VI.Vidi.213.150. Simon Weber is supported by the Swiss National Science Foundation under project no. 204320.

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
