# Supplemental:
# Training Fully Connected Neural Networks is $\exists\mathbb{R}$-Complete

## A $\quad\exists\mathbb{R}$-Membership

Membership in $\exists\mathbb{R}$ is already proven by Abrahamsen, Kleist and Miltzow in [3]. For the sake of completeness, while not being too repetitive, we shortly summarize their argument.

Applying a theorem from Erickson, van der Hoog and Miltzow [33], $\exists\mathbb{R}$-membership can be shown by describing a polynomial-time *real verification algorithm*. Such an algorithm gets a TRAIN-F2NN instance $I$ as well as a certificate $\Theta$ consisting of real-valued weights and biases as its input. The instance $I$ consists of a set $D$ of data points, a network architecture and a target error $\gamma$. The algorithm then needs to verify that the neural network described by $\Theta$ fits all data points in $D$ with a total error at most $\gamma$. The real verification algorithm is executed on a real RAM (see [33] for a formal definition). Thus, $\exists\mathbb{R}$-membership can be shown just like NP-membership, the main difference being the underlying machine model (for NP-membership the verification algorithm must run on a word RAM instead).

A real verification algorithm for TRAIN-F2NN loops over all data points in $D$ and evaluates the function described by the neural network for each of them. As in our case each hidden neuron uses $\mathrm{ReLU}$ as its activation function, each such evaluation takes linear time in the size of the network. The loss function can be computed in polynomial time on the real RAM (see also Definition 2 and the text afterwards).

## B $\quad\exists\mathbb{R}$-Hardness

In this section we present our $\exists\mathbb{R}$-hardness reduction for TRAIN-F2NN. The reduction is mostly geometric, so we start by reviewing the underlying geometry of the two-layer neural networks considered in the paper in Appendix B.1. This is followed by a high-level overview of the reduction in Appendix B.2 before we describe the gadgets in detail in Appendix B.3. Finally, in Appendix B.4, we combine the gadgets into the proof of Theorem 3.

### B.1 Geometry of Two-Layer Neural Networks

Our reduction below outputs a TRAIN-F2NN instance for a fully connected two-layer neural network $N$ with two input neurons, two output neurons, and $m$ hidden neurons. As defined above, for given weights and biases $\Theta$, the network $N$ realizes a function $f(\cdot, \Theta) : \mathbb{R}^2 \to \mathbb{R}^2$. The goal of this appendix is to build a geometric understanding of $f(\cdot, \Theta)$. We point the interested reader to these articles [6, 26, 49, 66, 92] investigating the set of functions exactly represented by different architectures of ReLU networks.

The $i$-th hidden $\mathrm{ReLU}$ neuron $v_i$ realizes a function

$$f_i : \mathbb{R}^2 \to \mathbb{R}$$
$$(x_1, x_2) \mapsto \mathrm{ReLU}(a_{1,i}x_1 + a_{2,i}x_2 + b_i),$$

where $a_{1,i}$ and $a_{2,i}$ are the edge weights from the first and second input neuron to $v_i$ and $b_i$ is its bias. We see that $f_i$ is a continuous piecewise linear function: If $a_{1,i} = a_{2,i} = 0$, then $f_i = \max\{b_i, 0\}$ everywhere. Otherwise, the domain $\mathbb{R}^2$ is partitioned into two half-planes, touching along a so-called *breakline* given by the equation $a_{1,i}x_1 + a_{2,i}x_2 + b_i = 0$. The two half-planes are (see Figure 6)

- the *inactive region* $\{(x_1, x_2) \subseteq \mathbb{R}^2 \mid a_{1,i}x_1 + a_{2,i}x_2 + b_i \leq 0\}$ in which $f_i$ is constantly 0, and
- the *active region* $\{(x_1, x_2) \subseteq \mathbb{R}^2 \mid a_{1,i}x_1 + a_{2,i}x_2 + b_i > 0\}$ in which $f_i$ is positive and has a constant gradient.

Now let $c_{i,1}, c_{i,2}$ be the weights of the edges connecting $v_i$ with the first and second output neuron, respectively, and let $f(\cdot, \Theta) = (f^1(\cdot, \Theta), f^2(\cdot, \Theta))$. For $j \in \{1, 2\}$, the function $f^j(\cdot, \Theta) = \sum_{i=1}^m c_{i,j} \cdot f_i(\cdot, \Theta)$ is a weighted linear combination of the functions computed at the hidden neurons. We make three observations:

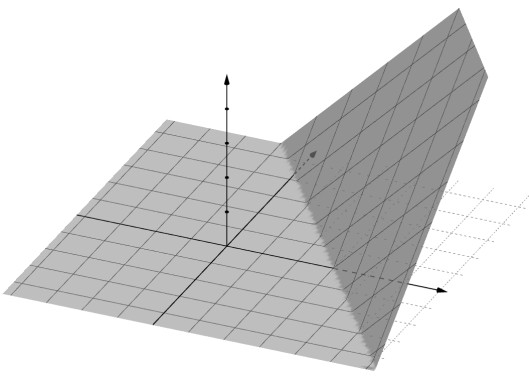

Figure 6: A continuous piecewise linear function computed by a hidden $\mathrm{ReLU}$ neuron. It has exactly one breakline; the left flat part is called the inactive region, whereas the right sloped part is the active region.

- As each function computed by a hidden $\mathrm{ReLU}$ neuron has at most one breakline, the domain of $f^j(\cdot, \Theta)$ is partitioned into the cells of a line arrangement of at most $m$ *breaklines*. Inside each of these cells $f^j(\cdot, \Theta)$ has a constant gradient.
- The position of the breakline that a hidden neuron $v_i$ contributes to $f^j(\cdot, \Theta)$ is determined solely by the $a_{\cdot,i}$ and $b_i$. In particular it is independent of $c_{i,j}$. Thus, the sets of breaklines partitioning $f^1(\cdot, \Theta)$ and $f^2(\cdot, \Theta)$ are both subsets of the same set: the set of (at most $m$) breaklines determined by the hidden neurons.
- Even if all $m$ hidden neurons compute a function with a breakline, $f^j(\cdot, \Theta)$ might have fewer breaklines: It is possible for a breakline to be *erased* by setting $c_{i,j} = 0$, or for breaklines created by different hidden neurons to cancel each other out (producing no breakline) or lie on top of each other (combining multiple breaklines into one). In our reduction, we make use of $c_{i,j} = 0$ to erase some breaklines in a single output dimension, but we avoid the other two cases of breaklines combining/canceling.

Note that these observations even imply another, stronger, statement: For each breakline, the change of the gradient of $f(\cdot, \Theta)$ when crossing the line is constant along the whole line (see also [26]). This allows to distinguish the following *types* of breaklines, which will ease our argumentation later.

**Definition 5.** A breakline $\ell$ is *concave* (*convex*) in $f^j(\cdot, \Theta)$, if the restriction of $f^j(\cdot, \Theta)$ to any two cells separated by $\ell$ in the breakline arrangement is concave (convex).

The *type* of a breakline is a tuple $(t_1, t_2) \in \{\wedge, 0, \vee\}^2$ describing whether the breakline is concave ($\wedge$), erased (0), or convex ($\vee$) in $f^1(\cdot, \Theta)$ and $f^2(\cdot, \Theta)$, respectively.

We have now established a basic geometric understanding of the function $f(\cdot, \Theta)$ computed by the neural network $N$. In our reduction we construct a data set which can be fit by a continuous piecewise linear function with $m$ breaklines if and only if a given ETR-INV instance has a solution. To make sure that the continuous piecewise linear function translates to a solution of the constructed TRAIN-F2NN instance, we need the following observation.

**Observation 6.** *Let $f : \mathbb{R}^2 \to \mathbb{R}^2$ be a continuous piecewise linear function that can be described via a line arrangement $\mathcal{L}$ of $m$ lines with the following properties:*

- *In at least one cell of $\mathcal{L}$ the value of $f$ is constantly $(0, 0)$.*
- *For each line $\ell \in \mathcal{L}$ the change of the gradient of $f$ when crossing $\ell$ is constant along $\ell$.*

*Then there is a fully connected two-layer neural network with $m$ hidden neurons computing $f$.*

To see that this observation is true, consider the following construction. For each breakline add a hidden neuron realizing the breakline with the inactive region towards the constant-$(0, 0)$ cell, and with the correct change of gradients in each output dimension. It is easy to see that the sum of all these neurons computes $f$. For a precise characterization of the functions representable by 2-layer neural networks with $m$ hidden neurons, we refer to [26].

## B.2 Reduction Overview

We show $\exists\mathbb{R}$-hardness of TRAIN-F2NN by giving a polynomial-time reduction from ETR-INV to TRAIN-F2NN. ETR-INV is a variant of ETR that is frequently used as a starting point for $\exists\mathbb{R}$-hardness proofs in the literature [2, 59, 30, 3].

Formally, ETR-INV is a special case of ETR in which the quantifier-free part $\varphi$ of the input sentence $\Phi \equiv \exists X_1, \ldots, X_n \in \mathbb{R} : \varphi(X_1, \ldots, X_n)$ is a conjunction (only $\wedge$ is allowed) of constraints, each of which is either of the form $X + Y = Z$ or $X \cdot Y = 1$. Further, $\Phi$ either has no solution or one with all values in $[1/2, 2]$.

**Theorem 7** ([2, Theorem 3.2]). ETR-INV *is* $\exists\mathbb{R}$-*complete.*

Furthermore, ETR-INV exhibits the same algebraic universality we seek for TRAIN-F2NN:

**Theorem 8** ([4]). *Let $\alpha$ be an algebraic number. Then there exists an instance of* ETR-INV*, which has a solution when the variables are restricted to $\mathbb{Q}[\alpha]$, but no solution when the variables are restricted to a field $\mathbb{F}$ that does not contain $\alpha$.*

The reduction starts with an ETR-INV instance $\Phi$ and outputs an integer $m$ and a set of $n$ data points such that there is a fully connected two-layer neural network $N$ with $m$ hidden neurons exactly fitting all data points ($\gamma = 0$) if and only if $\Phi$ is true. Recall that for fixed weights and biases $\Theta$ the neural network $N$ defines a continuous piecewise linear function $f(\cdot, \Theta) : \mathbb{R}^2 \to \mathbb{R}^2$.

For the reduction we define several *gadgets* representing the variables as well as the linear and inversion constraints of the ETR-INV instance $\Phi$. Strictly speaking, a gadget is defined by a set of data points that need to be fit exactly. These data points serve two tasks: Firstly, most of the data points are used to enforce that $f(\cdot, \Theta)$ has $m$ breaklines with predefined orientations and at almost predefined positions. Secondly, the remaining data points enforce relationships between the exact positions of different breaklines.

Globally, our construction yields $f(x, \Theta) = (0, 0)$ for "most" $x \in \mathbb{R}^2$. Each gadget consists of a constant number of parallel breaklines (enforced by data points) that lie in a *stripe* of constant width in $\mathbb{R}^2$. Only within these stripes $f(\cdot, \Theta)$ possibly attains non-zero values. Where two or more of these stripes intersect, additional data points can encode relations between the gadgets. The semantic meaning of a gadget is fully determined by the distances between its parallel breaklines. Thus each gadget can be translated and rotated arbitrarily without affecting its meaning.

**Simplifications.** Describing all gadgets purely by their data points is tedious and obscures the relatively simple geometry enforced by these data points. We therefore introduce two additional constructs, namely *data lines* and *weak data points*, that simplify the presentation. In particular, data lines impose breaklines, which in turn are needed to define gadgets. Weak data points are there to ensure that the gadgets used in the reduction encode variables with bounded range and that we can have features that are only active in one output dimension. How these constructs can be realized with carefully placed data points is deferred to Appendices B.3.5 and B.3.6.

- A data line $(\ell; y)$ consists of a line $\ell \subseteq \mathbb{R}^2$ and a label giving the ground truth value $y \in \mathbb{R}^2$. When describing a single gadget, we want that all points $p \in \ell$ are exactly fit, that is, $f(p, \Theta) = y$. As soon as we consider several gadgets, their corresponding stripes in $\mathbb{R}^2$ might intersect and we do not require that the data lines are fit correctly inside these intersections. As each data line will be enforced by placing finitely many data points on it, we choose coordinates for these defining data points that do not lie in any of the intersections.
- A weak data point relaxes a regular data point and prescribes only a lower bound on the value of the label. For example, we denote by $(x; y_1, \geq y_2)$ that $f^1(x, \Theta) = y_1$, and $f^2(x, \Theta) \geq y_2$. Weak data points can have such an inequality label in the first, the second, or both output dimensions.

## B.3 Gadgets and Constraints

We describe all gadgets in isolation first and consider the interaction of two or more gadgets only where it is necessary. In particular, we assume that $f(x, \Theta)$ is constantly zero for $x \in \mathbb{R}^2$ outside of the outermost breaklines enforced by each gadget. After all gadgets have been introduced, we

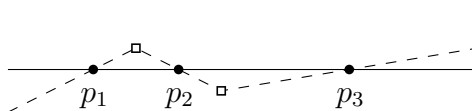
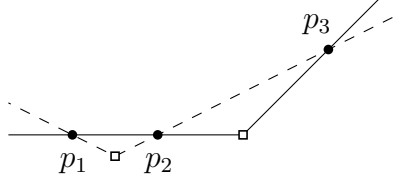

(a) If the points are collinear, then there is either no breakpoint or there are at least two.

(b) If the points are not collinear, then we need a breakpoint of a certain type, here convex.

Figure 7: Three consecutive points $p_1$, $p_2$ and $p_3$ in a cross section and possible interpolations through them (solid and dashed).

describe the global arrangement of the gadgets in Appendix B.4. Recall that, since each gadget can be freely translated and rotated, we can describe the positions of all its data lines and (weak) data points relative to each other.

Not all gadgets make use of the two output dimensions. Some gadgets have the same labels in both output dimensions for all of their data lines, and thus look the same in both output dimensions. For these gadgets we simplify the usual notation of $(y_1, y_2) \in \mathbb{R}^2$ for labels to single-valued labels $y \in \mathbb{R}$. In our figures, data points and functions which look the same in both output dimensions are drawn in black, while features only occurring in one dimension are drawn in orange and blue to distinguish the dimensions from each other.

Let $(\ell_1, y_i), \dots, (\ell_k, y_k)$ be parallel data lines describing (parts of) a gadget and $\ell \subseteq \mathbb{R}^2$ be an oriented line intersecting all $\ell_i$. Without loss of generality, we assume that the $\ell_i$ are numbered such that $\ell_i$ intersects $\ell$ before $\ell_j$ if and only if $i < j$. Then, this defines a *cross section* through the gadget: Formally, for each data line $(\ell_i, y_i)$ the cross section contains a point $p_i = (x_i, y_i) \in \mathbb{R} \times \mathbb{R}^2$, where $x_i$ is the oriented distance between the intersections of $\ell_1$ and $\ell_i$ on $\ell$. We say that two points $p_i$ and $p_j$ in the cross section are *consecutive*, if $|i - j| = 1$. If $\ell$ is perpendicular to all $\ell_i$, then the cross section is *orthogonal*.

Each $p_i = (x_i, y_i)$ in a cross section is a point in $\mathbb{R} \times \mathbb{R}^2$. When drawing a cross section in the following figures, we project it into a 2-dimensional coordinate system by marking $x_i$ along the abscissa and $y_i$ along the ordinate; if a $y_i$ behaves differently in the two output dimensions, we draw it twice distinguishing the two dimensions by color (and clearly marking which two drawn points belong together).

**Observation 9.** *For a single output dimension, whenever three consecutive points $p_i, p_{i+1}, p_{i+2}$ are not collinear in the cross section, then there must be a* breakpoint *(the intersection of a breakline $b$ with the cross section) strictly between $p_i$ and $p_{i+2}$. Further, if $p_{i+2}$ is above the line through $p_i$ and $p_{i+1}$, $b$ must be convex ($\vee$) in this output dimension. If otherwise $p_{i+2}$ is below that line, $b$ must be concave ($\wedge$) in this output dimension.*

**Observation 10.** *Whenever three consecutive points $p_1, p_{i+1}, p_{i+2}$ are collinear in one of the two output dimensions, then there is either no breakpoint (active in this dimension) strictly between $p_i$ and $p_{i+2}$, or at least two.*

Observations 9 and 10 are illustrated in Figure 7. In the analysis of each gadget, we use these observations to prove that the data lines enforce breaklines of a certain type with a prescribed orientation and (almost) fixed position.

### B.3.1 Variable Gadget

Recall that an instance of ETR-INV can be assumed to either have a solution with all variables in $[1/2, 2]$ or to have no solution at all. As already motivated in Section 6, variables are encoded as the slope of $f(\cdot, \Theta)$ at certain points. Recall that the gadget only affects a stripe of bounded width. See Figure 8 for a cross section view through this stripe.

A variable gadget consists of four parallel breaklines $b_1, b_2, b_3, b_4$, numbered from left to right. Left of $b_1$ and right of $b_4$ the value of $f(\cdot, \Theta)$ is constantly 0. Between $b_2$ and $b_3$ the value of $f(\cdot, \Theta)$ is constantly 6. The gradient of $f(\cdot, \Theta)$ between $b_1$ and $b_2$ is orthogonal to the breaklines and oriented towards $b_2$. We call the Euclidean norm of the gradient between $b_1$ and $b_2$ the *slope* of the variable

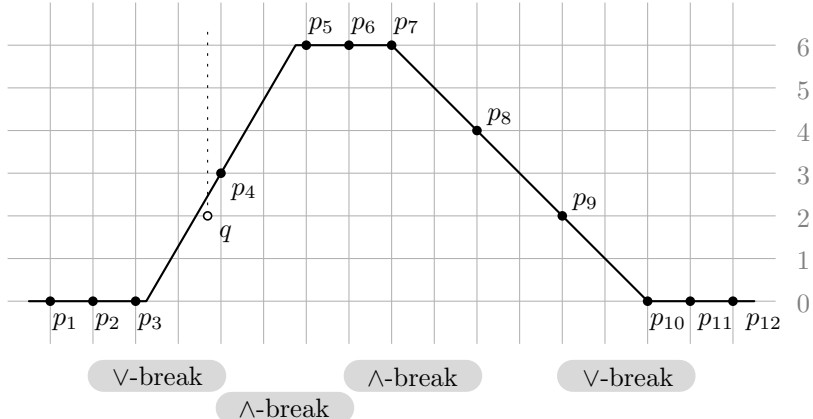

Figure 8: Orthogonal cross section view of a variable gadget. The black points $p_i$ are the projections of the data lines $\ell_1, \ldots, \ell_{12}$, the white point $q$ is a weak data point imposing a lower bound on $f(q, \Theta)$. The bars below the cross section indicate non-collinear triples, where convex ($\vee$) and concave ($\wedge$) breakpoints are needed. For example, there needs to be a convex breakpoint between $p_2$ and $p_4$.

gadget. The slope $s_X$ of a variable gadget for a variable $X$ is at least $3/2$ (if $b_1 = \ell_3$ and $b_2 = \ell_5$) and at most 3 (if $f(\cdot, \Theta)$ goes through the lowest possible value for the weak data point $q$). In order to represent values in $[1/2, 2]$ we say that a slope $s_X$ encodes the value $X = s_X - 1$.

The gradient between $b_3$ and $b_4$ carries no semantic meaning. It is merely used to bring $f(\cdot, \Theta)$ back to 0, such that only a stripe in $\mathbb{R}^2$ is affected by the gadget. This part is thus fixed to a slope of 1 for simplicity.

We realize a variable gadget using twelve parallel data lines, as described in the following table:

| | $\ell_1$ | $\ell_2$ | $\ell_3$ | $\ell_4$ | $\ell_5$ | $\ell_6$ | $\ell_7$ | $\ell_8$ | $\ell_9$ | $\ell_{10}$ | $\ell_{11}$ | $\ell_{12}$ |
|---|---|---|---|---|---|---|---|---|---|---|---|---|
| distance to $\ell_1$ | 0 | 1 | 2 | 4 | 6 | 7 | 8 | 10 | 12 | 14 | 15 | 16 |
| label | 0 | 0 | 0 | 3 | 6 | 6 | 6 | 4 | 2 | 0 | 0 | 0 |

**Lemma 11.** *Assume that at most four breaklines may be used. Then the twelve parallel data lines $\ell_1, \ldots, \ell_{12}$ as described in the table above realize a variable gadget with slope in $[3/2, \infty)$, thus carrying a value in $[1/2, \infty)$.*

*Proof.* We first prove that four breaklines are necessary to fit all data lines exactly. For this, consider an orthogonal cross section through the data lines. It is easy to see that the variable gadget has four non-collinear triples (see Figure 8) and that they pairwise share at most one point. Thus, by Observation 9, four breaklines $b_1, b_2, b_3, b_4$ are indeed required.

As $p_1, p_2, p_3$ are collinear, we can further conclude by Observation 10 that $b_1$ has to intersect the cross section at $p_3$ or strictly between $p_3$ and $p_4$. Similarly, as $p_5, p_6, p_7$ are collinear, we get that $b_2$ has to intersect the cross section at $p_5$ or strictly between $p_4$ and $p_5$. The remaining breaklines $b_3$ and $b_4$ can only intersect the cross section on $p_7$ and $p_{10}$, respectively. Since this holds at every orthogonal cross section through the data lines, we further conclude that the breaklines are parallel to each other and to the data lines.

The exact positions of $b_1$ and $b_2$ depend on each other. As $f(\cdot, \Theta)$ must fit $\ell_4$, the distance between $\ell_4$ and $b_1$ equals the distance between $\ell_4$ and $b_2$. If $b_1 = \ell_3$ and $b_2 = \ell_5$, the slope of $f(\cdot, \Theta)$ between $b_1$ and $b_2$ is exactly $3/2$. This is the minimum possible slope, because $\ell_3$ and $\ell_5$ have to be fit. There is no restriction on the maximum possible slope. Thus $\ell_1, \ldots, \ell_{12}$ realize a variable gadget carrying a value in $[1/2, \infty)$. $\square$

It remains to bound the value of the variable also from above, such that it is constrained to the interval $[1/2, 2]$. To achieve this, we use a weak data point, named $q$ in Figure 8. Recall that the label of a weak data point is a lower bound to $f(\cdot, \Theta)$.

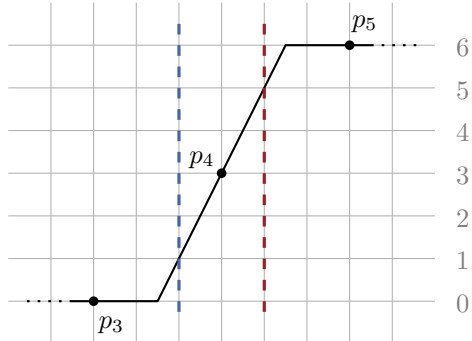

Figure 9: Partial cross section of a variable gadget with slope $s_X$ showing the lower (blue) and upper (red) measuring lines. The contribution of this variable gadget at these lines is $3 - s_X$ and $3 + s_X$, respectively. As can be seen, this variable gadget has a slope of $s_X = 2$, thus encoding $X = 1$.

**Lemma 12.** *Let $q$ be a weak data point at distance $3 + 2/3$ to $\ell_1$ (and thus a distance of $1/3$ to $\ell_4$) with lower bound label $\geq 2$. Then the slope of the variable gadget is at most $3$.*

*Proof.* Assume for the sake of contradiction that the slope of $f(\cdot, \Theta)$ between $b_1$ and $b_2$ is strictly larger than $3$. Then the contribution of the variable gadget to $f(q, \Theta)$ is strictly less than $2$, so the lower bound label of $q$ is not satisfied, a contradiction. $\qquad\square$

We conclude that with twelve data lines and one weak data point we can enforce four parallel breaklines forming a variable gadget, with a minimum slope of $3/2$ and a maximum slope of $3$, thus encoding a value in the interval $[1/2, 2]$.

### B.3.2 Measuring a Value from a Variable Gadget

For every particular variable gadget we consider the two parallel lines with distance $1$ to $\ell_4$ to be its *measuring lines*. We distinguish the *lower measuring line* (the one towards $\ell_3$) and the *upper measuring line* (the one towards $\ell_5$). Note that, since the slope of the variable gadget is restricted to be in the interval $[3/2, 3]$, both measuring lines are always inside or at the boundary of the sloped part (in other words, between breaklines $b_1$ and $b_2$), see Figure 9.

Let again $s_X$ be the slope of the variable gadget carrying the value of variable $X$. At any point $p$ on $\ell_4$, the contribution of the variable gadget to $f(p, \Theta)$ is exactly $3$, assuming $\ell_4$ is fit exactly. From this it follows that for a point $p_u$ on the upper measuring line the contribution to $f(p_u, \Theta)$ is $3 + s_X$. Thus, if we know the value of $f(p_u, \Theta)$ and further that $p_u$ belongs to a single variable gadget only, then we get $X = s_X - 1 = f(p_u, \Theta) - 4$ for the value represented by the variable gadget. Similarly, for a point $p_l$ on the lower measuring line the contribution to $f(p_l, \Theta)$ is $3 - s_X$. If $p_l$ belongs to a single variable gadget only, then $X = s_X - 1 = 2 - f(p_l, \Theta)$ is the represented value.

### B.3.3 Enforcing Linear Constraints between Variables: Addition and Copying

Until now we always only considered one gadget in isolation. As soon as we have two or more gadgets, their corresponding stripes may intersect. Inside these intersections, the different gadgets interfere and below we describe how to use this interference to encode (non-)linear constraints. Let us note, however, that data lines are not fit correctly inside these intersections any more. This is not a problem because each data line is later replaced by just three collinear data points outside of any intersections of stripes, see Appendix B.3.6 below. As we will see there, it is enough that these three data points are fit exactly.

For disjoint subsets $\mathcal{A}$ and $\mathcal{B}$ of the variables we can use an additional data point $p$ to enforce a linear constraint of the form $\sum_{A \in \mathcal{A}} A = \sum_{B \in \mathcal{B}} B$. Note that this type of constraint in particular allows us to copy a value from one variable gadget to another ($X = Y$ via $\mathcal{A} = \{X\}, \mathcal{B} = \{Y\}$) or to encode addition ($X + Y = Z$ via $\mathcal{A} = \{X, Y\}, \mathcal{B} = \{Z\}$).

The data point $p$ is placed on a measuring line of each involved variable. For all variables in $\mathcal{A}$ the data point $p$ must be on the upper measuring line of the corresponding variable gadget. Similarly, for variables in $\mathcal{B}$ the data point $p$ must be on the lower measuring line. Therefore the variable gadgets of the involved variables need to be positioned such that the needed measuring lines all intersect at a common point, where $p$ can be placed. This is trivial for $|\mathcal{A}| + |\mathcal{B}| \leq 2$, see Figure 10; but more involved for more variables, see Figure 11. Using the equality constraint $X = Y$, we can copies the value of a variable onto multiple variable gadgets, which can be positioned freely to obtain the required intersections. We discuss the global layout to achieve this in more detail in Appendix B.4.

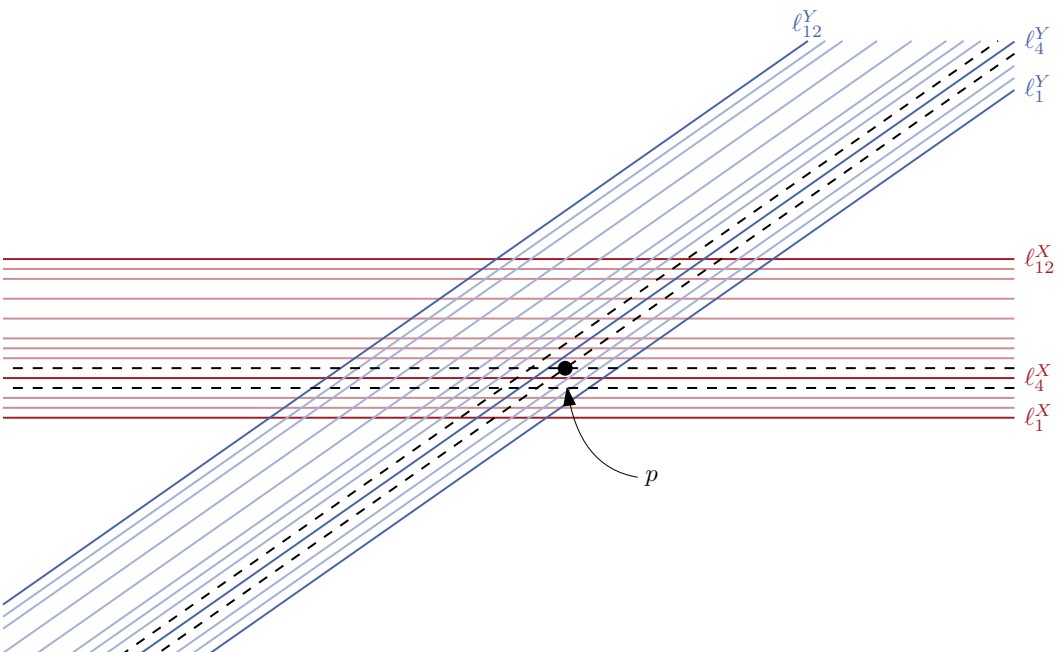

Figure 10: Top-down view on the intersection of two variable gadgets corresponding to two variables $X$ (red) and $Y$ (blue). The dashed lines are their measuring lines. The point $p$ is placed at the intersection of the upper measuring line for $X$ and lower measuring line for $Y$, and receives label 6 to enforce the constraint $X = Y$.

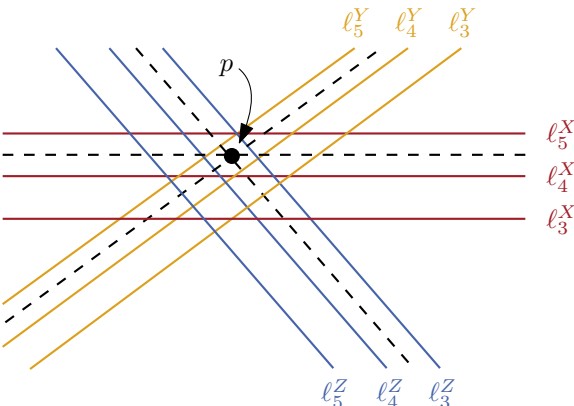

Figure 11: Top-down view of the "interesting part" of the intersection of three variable gadgets corresponding to variables $X$ (red), $Y$ (orange), and $Z$ (blue). The dashed lines are upper the measuring lines for $X$ and $Y$ and the lower measuring line for $Z$, intersecting in a single point $p$ with label 10. This realizes the constraint $X + Y = Z$.

**Lemma 13.** *The constraint $\sum_{A \in \mathcal{A}} A = \sum_{B \in \mathcal{B}} B$ can be enforced by a data point $p$ placed as described above with label $y = 4|\mathcal{A}| + 2|\mathcal{B}|$.*

*Proof.* First, let us consider a variable $A \in \mathcal{A}$ and let $s_A$ be the slope of the corresponding variable gadget. Data point $p$ is placed on the upper measuring line of the variable gadget, so it contributes $3 + s_A$ to $f(p, \Theta)$. Similarly, for a variable $B \in \mathcal{B}$ let $s_B$ be the slope of its corresponding variable gadget. Here $p$ is placed on the lower measuring line and this variable gadget contributes $3 - s_B$ to $f(p, \Theta)$.

The overall contribution of the variable gadgets of all involved variables adds up to

$$f(p, \Theta) = \sum_{A \in \mathcal{A}} (3 + s_A) + \sum_{B \in \mathcal{B}} (3 - s_B)$$
$$= \sum_{A \in \mathcal{A}} (4 + A) + \sum_{B \in \mathcal{B}} (2 - B),$$

where we used that the value represented by a variable gadget is its slope minus 1. Choose the label of $p$ to be $y = 4|\mathcal{A}| + 2|\mathcal{B}|$. Then, $p$ is fit exactly if and only if the linear constraint $\sum_{A \in \mathcal{A}} A = \sum_{B \in \mathcal{B}} B$ is satisfied. $\square$

Lemma 13 is more general than we actually require it for our reduction. The only linear constraint in an instance of ETR-INV is the addition $X + Y = Z$. In our reduction we also need the ability to copy values, i.e., a constraint of the form $X = Y$. These are the only linear constraints required and can be encoded with data points using only two different labels:

**Observation 14.** *To encode the addition constraint $X + Y = Z$ of* ETR-INV *the data point has label* 10. *For the copy constraint $X = Y$ the data point has label* 6.

Until now we never distinguished between the two output dimensions of variable gadgets. Let us note here that it is enough for the data point enforcing the linear constraint to be active in only one output dimension. This holds, because each variable gadget has exactly the same breaklines (and therefore represents the same value) in both output dimensions. In particular, it may also be a weak data point with a lower bound label of $\geq 0$ in the other output dimension. This relaxation is used to realize inversion constraints in the following section.

### B.3.4 Inversion Gadget

We introduce an *inversion gadget* which is in essence the superposition of two variable gadgets. This gadget uses data lines with different labels in the two output dimensions to enforce the presence of five parallel breaklines $b_1, \ldots, b_5$, instead of the usual four for a normal variable gadget, see Figure 12. In each output dimension, the continuous piecewise linear function defined by these five breaklines looks like a normal variable gadget, so four of the five breaklines are required. This yields that the two variable gadgets in the two output dimensions share three breaklines and have one exclusive breakline each (which is erased in the other output dimension). As for a normal variable gadget, the last two breaklines ($b_4$ and $b_5$) are just there to bring $f(\cdot, \Theta)$ back to 0 outside of the stripe of the inversion gadget. In the first output dimension the slope between $b_1$ and $b_2$ encodes the value while $b_3$ is erased. Similarly, in the second output dimension the sloped between $b_2$ and $b_3$ encodes the value while $b_1$ is erased. Note that $b_2$ is involved in both sloped parts and therefore changing the value encoded by the variable gadget in one dimension has a directed effect on the value encoded by the variable gadget in the other dimension. As we will see, with carefully placed data lines this dependency encodes an inversion constraint.

We define an inversion gadget using 13 parallel data lines, positioned relatively to each other as in the following table:

| | $\ell_1$ | $\ell_2$ | $\ell_3$ | $\ell_4$ | $\ell_5$ | $\ell_6$ | $\ell_7$ | $\ell_8$ | $\ell_9$ | $\ell_{10}$ | $\ell_{11}$ | $\ell_{12}$ | $\ell_{13}$ |
|---|---|---|---|---|---|---|---|---|---|---|---|---|---|
| distance to $\ell_1$ | 0 | 1 | 2 | 4 | 7 | 9 | 10 | 11 | 13 | 15 | 17 | 18 | 19 |
| label in dim. 1 | 0 | 0 | 0 | 3 | 6 | 6 | 6 | 6 | 4 | 2 | 0 | 0 | 0 |
| label in dim. 2 | 0 | 0 | 0 | 0 | 3 | 6 | 6 | 6 | 4 | 2 | 0 | 0 | 0 |

**Lemma 15.** *Assume that at most five breaklines may be used. Then the* 13 *parallel data lines as described in the table above realize an inversion gadget carrying two values $X$ and $Y$ satisfying $X \cdot Y = 1$.*

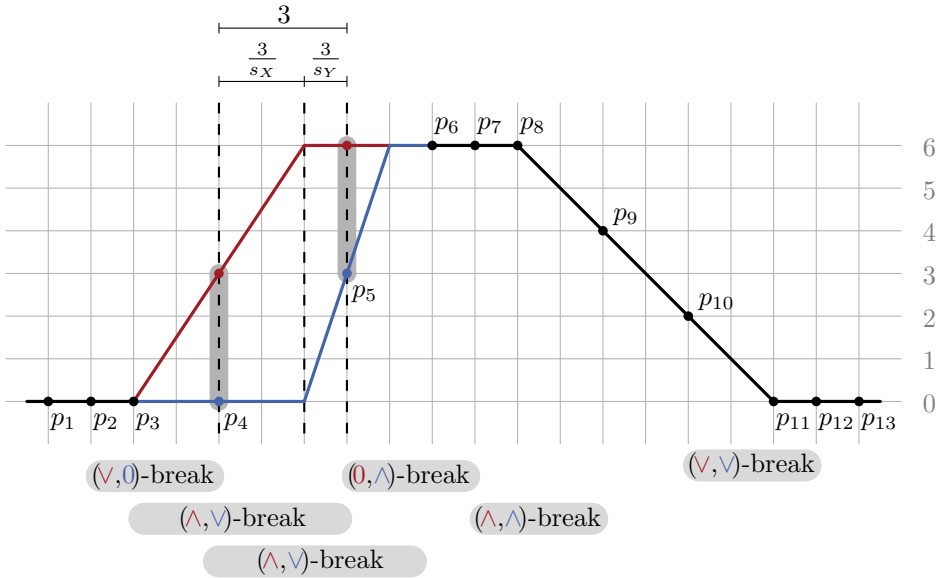

Figure 12: Cross section view of the inversion gadget. The points $p_1, \ldots, p_{13}$ are the projections of the data lines $\ell_1, \ldots, \ell_{13}$. Points $p_4$ and $p_5$ have different labels in the first (red) and second (blue) output dimension. Non-collinear triples of points force breaklines in-between them.

*Proof.* Again, we start by showing that at least five breaklines are necessary. First off, there must again be the two fixed breaklines $b_4$ and $b_5$ on the data lines $\ell_8$ and $\ell_{11}$, by the same arguments as for the normal variable gadget (see proof of Lemma 11). There are four more relevant triples of non-collinear points, requiring the following breaklines:

- Triple $p_2, p_3, p_4$ enforces a $(\vee, 0)$-breakline between $\ell_2$ and $\ell_4$. Actually, since $p_1, p_2, p_3$ are collinear, the breakline must be between $\ell_3$ and $\ell_4$.

- Triple $p_3, p_4, p_5$ enforces a $(\wedge, \vee)$-breakline between $\ell_3$ and $\ell_5$.

- Triple $p_4, p_5, p_6$ enforces a $(\wedge, \vee)$-breakline between $\ell_4$ and $\ell_6$.

- Triple $p_5, p_6, p_7$ enforces a $(0, \wedge)$-breakline between $\ell_5$ and $\ell_7$. Actually, since $p_6, p_7, p_8$ are collinear, the breakline must be between $p_6$ and $p_7$.

We need to fulfill all four of these requirements with only three remaining breaklines. By looking at the types we see that this is only possible if the triples $p_3, p_4, p_5$ and $p_4, p_5, p_6$ enforce the same breakline. This breakline must then be between $\ell_4$ and $\ell_5$.

As we now know the locations and types of all breaklines, we can analyze the relationship between the values $X$ and $Y$ carried on the two variable gadgets. The distance between $\ell_4$ and $\ell_5$ is 3 by construction. This distance can be subdivided into the distance from $\ell_4$ to $b_2$, and from $b_2$ to $\ell_5$. Between $\ell_4$ and $b_2$ function $f^1(\cdot, \Theta)$ rises from 3 to 6, thus the distance between $\ell_4$ and $b_2$ must be $(6-3)/s_X$. Similarly, between $b_2$ and $\ell_5$ function $f^2(\cdot, \Theta)$ rises from 0 to 3, thus the distance between $b_2$ and $\ell_5$ must be $(3-0)/s_Y$. These distances add up to 3, and thus the gadget encodes the constraint $3/s_X + 3/s_Y = 3$, or equivalently $3s_Y + 3s_X = 3s_X s_Y$. Using that $X = s_X - 1$ and $Y = s_Y - 1$ we get

$$3(Y+1) + 3(X+1) = 3(X+1)(Y+1)$$

which is true if and only if

$$X \cdot Y = 1. \qquad \square$$

To encode an $X \cdot Y = 1$ constraint of ETR-INV, we first identify two normal variable gadgets carrying the variables $X$ and $Y$. Then the inversion gadget is placed so that it intersects both. In the intersection with the $X$-variable gadget we copy $X$ to the first dimension of the inversion gadget and in the intersection with the $Y$-variable gadget we copy $Y$ to the second dimension of the inversion

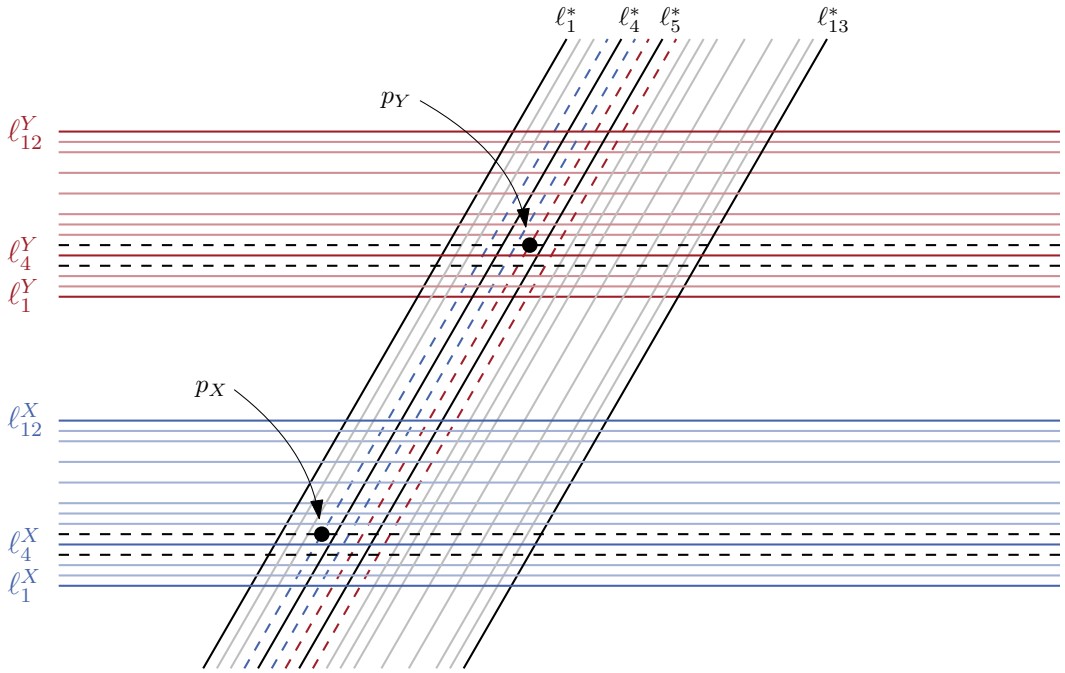

Figure 13: Top-down view on two variable gadgets (horizontal), denoted as $X$ (blue) and $Y$ (red) (the data lines are solid and the measuring lines are dashed). The sloped gadget is an inversion gadget linking the two variable gadgets. Two weak data points $p_X$ and $p_Y$ copy $X$ and $Y$ to the first and second dimension of the inversion gadget, respectively.

gadget. This copying can be done as described in Appendix B.3.3 using weak data points. See Figure 13 for a top-down view on this construction. As variable gadgets carry the same value in both output dimensions, enforcing the inversion constraint on just one dimension of each variable gadget ($X$ and $Y$) is sufficient.

Note that the sloped parts of an inversion gadget differ in the two output dimensions. In particular the measuring lines in the first dimension have distance 1 to $\ell_4$, while the measuring lines in the second dimension have distance 1 to $\ell_5$. Thus the weak data points used for copying need to be placed on the measuring lines of the correct dimension.

### B.3.5 Realizing Weak Data Points: Cancel Gadgets

Recall that, for our construction so far, we used weak data points with labels of the form $\geq y$ in one or both output dimensions. We now introduce a gadget which can be used to realize such a weak data point using only non-weak data points (with constant labels). For each weak data point $p$ we introduce a *cancel gadget*, consisting of three parallel breaklines that form a stripe containing $p$ but that shall not contain any other (weak) data point.

A cancel gadget can be active in either one of the two output dimensions, or in both of them. If a cancel gadget is active in some output dimension, the breaklines form a $\vee$ shape of variable height in that dimension. On the other hand, if the cancel gadget is inactive in some output dimension, the breaklines are all inactive (type 0) and the gadget contributes nothing to $f(\cdot, \Theta)$ in this dimension. The following table shows the locations and labels of the eight data lines that define a cancel gadget. The cancel gadget is illustrated in Figure 14.

|  | $\ell_1$ | $\ell_2$ | $\ell_3$ | $\ell_4$ | $\ell_5$ | $\ell_6$ | $\ell_7$ | $\ell_8$ |
|---|---|---|---|---|---|---|---|---|
| distance to $\ell_1$ | 0 | 1 | 2 | 3 | 5 | 6 | 7 | 8 |
| label in active dimension(s) | 0 | 0 | 0 | $-1$ | $-1$ | 0 | 0 | 0 |
| label in inactive dimension(s) | 0 | 0 | 0 | 0 | 0 | 0 | 0 | 0 |

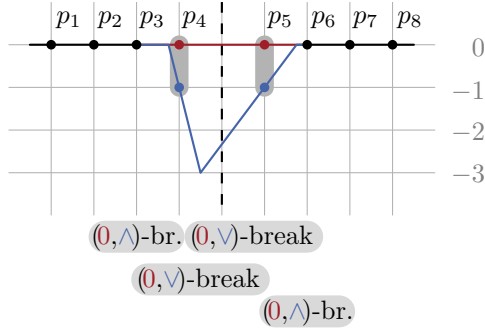
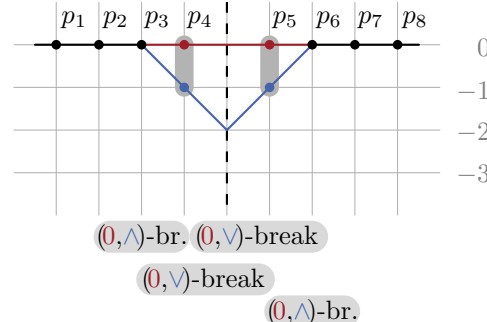

(a) The cancel gadget can be asymmetric.

(b) A cancel gadget has a maximum contribution to the weak data point of $-2$.

Figure 14: Cross sections of a cancel gadget which is inactive in the first (red) dimension and active in the second (blue) dimension. It is used to "cancel" a weak data point in the active dimension (blue) that lies on the dashed vertical line. In the inactive dimension (red) it does not contribute anything to $f(\cdot, \Theta)$, i.e., all breaklines are erased (type 0).

**Lemma 16.** *Assume that at most three breaklines may be used. The eight data lines as described above realize a cancel gadget that contributes $0$ to $f(p, \Theta)$ in an inactive dimension and an arbitrary amount $c \in (-\infty, -2]$ to $f(p, \Theta)$ in an active dimension to any point $p$ with equal distance to $\ell_4$ and $\ell_5$.*

*Proof.* Again we start by arguing that three breaklines are necessary in each active output dimension. There are the following non-collinear triples, each enforcing a breakline:

- Triple $p_2, p_3, p_4$ enforces a $\wedge$-breakline between $\ell_2$ and $\ell_4$. Actually, since $p_1, p_2, p_3$ are collinear, the breakline must be between $\ell_3$ and $\ell_4$.

- Triple $p_3, p_4, p_5$ enforces a $\vee$-breakline between $\ell_3$ and $\ell_5$.

- Triple $p_4, p_5, p_6$ enforces a $\vee$-breakline between $\ell_4$ and $\ell_6$.

- Triple $p_5, p_6, p_7$ enforces a $\wedge$-breakline between $\ell_5$ and $\ell_7$. Actually, since $p_6, p_7, p_8$ are collinear, the breakline must be between $\ell_5$ and $\ell_6$.

We see that the two $\wedge$-breaklines must be in disjoint intervals, so we indeed need two different breaklines. We have one breakline remaining to satisfy the two $\vee$-type enforcements. Thus we get $b_1$ of type $\wedge$ between $\ell_3$ and $\ell_4$, $b_2$ of type $\vee$ between $\ell_4$ and $\ell_5$, and $b_3$ of type $\wedge$ between $\ell_5$ and $\ell_6$. All breaklines must be inactive (type 0) in an inactive output dimension.

We can now analyze the possible contribution of the cancel gadget in an active output dimension to a point $p$ equidistant to $\ell_4$ and $\ell_5$. Any contribution $c \in (-\infty, -2]$ can be realized by placing breakline $b_1$ at distance $-1/(c+1)$ left of $\ell_4$, breakline $b_3$ at distance $-1/(c+1)$ right of $\ell_5$ and breakline $b_2$ equidistant between $\ell_4$ and $\ell_5$. A contribution $c > -2$ can not be realized, as otherwise there would need to be two convex breaklines, because $p_3, p_4, p$ and $p, p_5, p_6$ are both non-collinear triples requiring a convex breakline. $\qquad\square$

Since the breaklines are at the exact same positions in both output dimensions we can observe the following:

**Observation 17.** *If the cancel gadget is active in both output dimensions, it contributes the same amount to the weak data point in both dimensions.*

The cancel gadget is placed such that the weak data point is equidistant to $\ell_4$ and $\ell_5$. The inequality label $\geq y$ of the weak data point is converted into the constant label $y - 2$.

As shown above, the cancel gadget can contribute any value $c \in (-\infty, -2]$ to the data point. Thus, the data point can be fit perfectly if and only if the other gadgets contribute at least a value of $y$ to the data point, that is, the intended weak constraint is met.

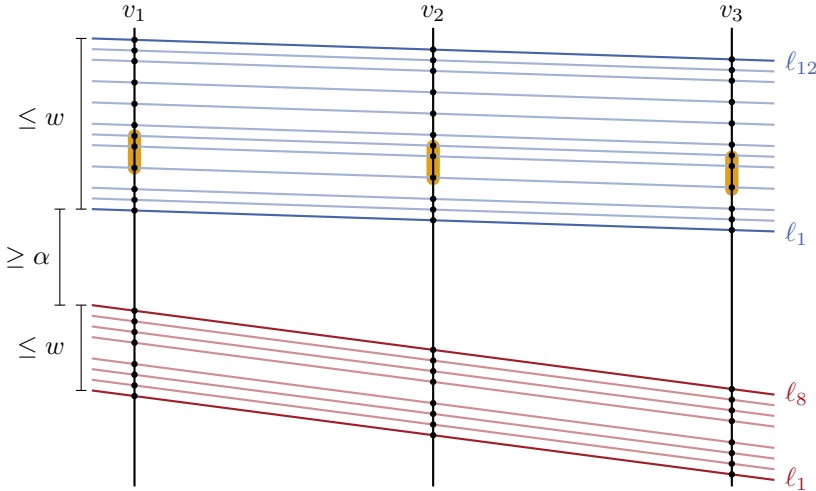

Figure 15: Data lines defining a variable gadget (blue) and a cancel gadget (red) and their intersections with the vertical lines $v_1, v_2, v_3$. We add a data point at each intersection. The values $\alpha$ and $w$ describe the minimal distance between data lines of different gadgets, and the maximal distance between data lines of the same gadget, respectively. In orange, we highlighted three matching breakpoint intervals (in this case forcing a $\wedge$-breakpoint between $\ell_4$ and $\ell_6$ of the variable gadget).

Lastly, let us note that a cancel gadget could also be constructed such that it has a $\wedge$ shape and contributes a positive value in $[2, \infty)$ to the data point. This would allow constraints of the form $\leq y$ as well. Combining two cancel gadgets, one positive and one negative, would even allow a data point to attain an arbitrary value in $\mathbb{R}$ (in one of the two dimensions), but this is not needed for our reduction.

### B.3.6 Realizing Data Lines using Data Points

We previously assumed that our gadgets are defined by data *lines*, but in the TRAIN-F2NN problem, we are only allowed to use data *points*. In this section, we argue that a set of data lines can be realized by replacing each data line by three data points. This in turn allows us to define the gadgets described throughout previous sections solely using data points. This section is devoted to showing the following lemma, which captures this transformation formally. Note that our replacement of data lines by data points does not work in full generality, but we show it for all the gadgets that we constructed.

**Lemma 18.** *Assume we are given a set of gadgets (variable gadgets, inversion gadgets and cancel gadgets), in total requiring $m$ breaklines. Further assume that the gadgets are placed in $\mathbb{R}^2$ such that no two parallel gadgets overlap. Then each data line can be replaced by three data points such that a continuous piecewise linear function with at most $m$ breaklines fits the data points if and only if it fits the data lines.*

For the proof consider the line arrangement induced by the data lines. We introduce three vertical lines $v_1, v_2, v_3$ to the right of all intersections between the data lines. The vertical lines are placed at unit distance to one another. In our construction, no data line is vertical, thus each data line intersects each of the vertical lines exactly once. We place one data point on each intersection of each vertical line with a data line. The new data point inherits the label of the underlying data line. Furthermore, on each vertical line, we ensure that the minimum distance $\alpha$ between any two data points belonging to different gadgets is larger than the maximum distance $w$ between data points belonging to the same gadget. This can be achieved by placing the $v_1, v_2$ and $v_3$ far enough to the right and by ensuring a minimum distance between parallel gadgets. See Figure 15 for an illustration.

Along each of the three vertical lines the data points form cross sections of all the gadgets, similar to the cross sections shown in Figures 8, 12 and 14 (note however that here the cross sections are not orthogonal). We have previously analyzed cross sections of individual gadgets in the proofs of Lemmas 11, 15 and 16. There we identified certain intervals between some of the data lines that need to contain a *breakpoint* (the intersection of a breakline and the cross section). If we now consider

Table 1: Location and type of the breaklines in variable gadgets, inversion gadgets, and cancel gadgets.

| | Location | Type |
|---|---|---|
| $b_1$ | $[\ell_3, \ell_4)$ | $(\vee, \vee)$ |
| $b_2$ | $(\ell_4, \ell_5]$ | $(\wedge, \wedge)$ |
| $b_3$ | on $\ell_7$ | $(\wedge, \wedge)$ |
| $b_4$ | on $\ell_{10}$ | $(\vee, \vee)$ |

(a) Variable gadget.

| | Location | Type |
|---|---|---|
| $b_1$ | $[\ell_3, \ell_4)$ | $(\vee, 0)$ |
| $b_2$ | $(\ell_4, \ell_5)$ | $(\wedge, \vee)$ |
| $b_3$ | $(\ell_5, \ell_6]$ | $(0, \wedge)$ |
| $b_4$ | on $\ell_8$ | $(\wedge, \wedge)$ |
| $b_5$ | on $\ell_{11}$ | $(\vee, \vee)$ |

(b) Inversion gadget.

| | Location | Type |
|---|---|---|
| $b_1$ | $[\ell_3, \ell_4)$ | $(0, \wedge)$ |
| $b_2$ | $(\ell_4, \ell_5)$ | $(0, \vee)$ |
| $b_3$ | $(\ell_5, \ell_6]$ | $(0, \wedge)$ |

(c) Cancel gadget.

the cross sections of all gadgets along the vertical lines, we refer to these intervals as *breakpoint intervals*. A breakpoint interval may degenerate to just one point (we have seen this for example for the fixed-slope side of a variable gadget). By our placement of the vertical lines, the cross sections (and thus also the breakpoint intervals) of different gadgets do not overlap.

Any two data lines bounding a breakpoint interval on $v_1$ also bound a breakpoint interval on $v_2$ and $v_3$. We call the three breakpoint intervals on $v_1$, $v_2$ and $v_3$ which are bounded by the same data lines *matching* breakpoint intervals.

In total there are $3m$ breakpoint intervals. We show that the only way to *stab* each of them exactly once using $m$ breaklines is if each breakline stabs exactly three matching breakpoint intervals. The first observation towards this is that each breakline can only stab a single breakline interval per vertical line because all breakline intervals are pairwise disjoint. Thus, having $m$ breakpoint intervals on each vertical line, each of the $m$ breaklines has to stab exactly three intervals, one per vertical line. In a first step, we show that each breakline has to stab three breakpoint intervals belonging to the same gadget.

**Claim 19.** *Each breakline has to stab three breakpoint intervals of the same gadget.*

*Proof.* The proof is by induction on the number of gadgets. For a single gadget the claim trivially holds. For the inductive step, we consider the lowest gadget $g$ (on $v_1$, $v_2$ and $v_3$) and assume for the sake of contradiction that there is a breakline $b$ stabbing a breakpoint interval of $g$ on $v_2$ and a breakpoint interval of a different gadget $g'$ above $g$ on $v_1$. By construction, the minimum distance $\alpha$ between different gadgets is larger than the maximum width $w$ of any gadget on all three vertical lines. Thus, the distance of any breakpoint interval of $g'$ to any breakpoint interval of $g$ on $v_1$ is larger than the width of $g$ on $v_3$. Therefore, we know that the breakline $b$ intersects $v_3$ below any breakpoint intervals of $g$, which is the lowest gadget on $v_3$. Thus it stabs at most two breakpoint intervals in total and therefore not all intervals can be stabbed. The same reasoning holds if the roles of $v_1$ and $v_3$ are flipped. All breaklines stabbing breakpoint intervals of $g$ on $v_2$ must therefore also stab breakpoint intervals of $g$ on $v_1$ and $v_3$. Applying the induction hypothesis on the remaining gadgets, it follows that each breakline only stabs breakpoint intervals of the same gadget. □

We can therefore analyze the situation for each gadget in isolation. The main underlying idea is to use the type of the required breakline. Each breaklines must stab three breakpoint intervals of the same type. Let us summarize the findings about required breakline locations and types from the proofs of Lemmas 11, 15 and 16 in Table 1.

**Claim 20.** *To stab all breakpoint intervals of a variable gadget with only four breaklines, each of them has to stab three matching breakpoint intervals.*

*Proof.* See Table 1a. On the three vertical lines, there are six breakpoint intervals for breaklines of type $(\vee, \vee)$ in total. If only two breaklines should stab these six breakpoint intervals, one breakline needs to stab at least two of the single-point intervals. If a breakline goes through two of the single points, it also goes through the third point, and can thus not go through the proper intervals. Therefore one breakline must stab the single-point intervals, and the other one stabs the proper breakpoint intervals.

The same argument can be made for the breakpoint intervals of type $(\wedge, \wedge)$, and thus each breakline stabs three matching breakpoint intervals. □

**Claim 21.** *To stab all breakpoint intervals of an inversion gadget with only five breaklines, each of them has to stab three matching breakpoint intervals.*

*Proof.* See Table 1b. All five sets of three matching breakpoint intervals have a different type of required breakline, thus each breakline stabs three matching breakpoint intervals. □

**Claim 22.** *To stab all breakpoint intervals of a cancel gadget with only three breaklines, each of them has to stab three matching breakpoint intervals.*

*Proof.* See Table 1c. There is only one set of three breakpoint intervals for a breakline of type $(0, \vee)$, so it is trivially matched correctly.

We can see that the breakpoint intervals for a breakline of type $(0, \wedge)$ have a distance of 2 from each other, and each have a width of 1. If the two breaklines of this type would not stab three matching breakpoint intervals, one of them would need to stab two matching intervals and one non-matching interval. As the distance between the vertical lines is equal and the breakpoint intervals are further apart from each other than their width, there is no way for a breakline to lie in this way. We conclude that all breaklines stab three matching breakpoint intervals. □

From Claims 20 to 22 it also follows that within a single gadget, between the vertical lines no two breaklines can cross each other, nor can they cross a data line. Together with Claim 19, we can finally prove Lemma 18.

*Proof of Lemma 18.* By Claim 19 it follows that every breakline must stab three breakpoint intervals of the same gadget. By Claims 20 to 22 it follows then that each breakline must stab three matching breakpoint intervals, and therefore the breaklines do not cross any data lines between the three vertical lines.

It remains to show that the data points already ensure that each breakline $b$ is parallel to the two parallel data lines $d$ and $d'$ enclosing it. To this end, consider the parallelogram defined by $d, d', v_1, v_3$ (see Figure 16) and let $j$ be an output dimension in which $b$ is active (not erased). Since no other breakline intersects this parallelogram, we obtain that $f^j$ has exactly two linear pieces within the parallelogram, which are separated by $b$. Moreover, since $b$ stabs matching breakpoint intervals, the three data points on $d$ must belong to one of the pieces. Since these points have the same label, it follows that the gradient of this piece in output dimension $j$ must be orthogonal to $d$ (and, thus, to $d'$ as well). Applying the same argument on the data points on $d'$, we obtain that the gradient of the other piece must be orthogonal to $d$ and $d'$ as well. This implies that also the difference of the gradients of the two pieces is orthogonal to $d$ and $d'$. Finally, since $b$ must be orthogonal to this difference of gradients, we obtain that it is parallel to $d$ and $d'$. □

## B.4   Global Construction Layout

We can now finalize proving $\exists\mathbb{R}$ hardness of TRAIN-F2NN For each variable $X$ of the ETR-INV formula, we build a horizontal *canonical* variable gadget carrying this variable at the bottom of the construction. As argued previously, the variable gadgets naturally ensure $X \geq 1/2$ for all variables $X$ and the added weak data point with label $\geq 2$ ensures $X \leq 2$.

The constraints of the form $X + Y = Z$ are enforced by copying the three involved variables onto three new variable gadgets. These three new variable gadgets are positioned to intersect above all horizontal variable gadgets in a way such that their correct measuring lines intersect in a single point. This allows a data point enforcing the constraint to be placed.

For the $k$ inversion constraints of the form $X \cdot Y = 1$ we build an array of $k$ parallel inversion gadgets intersecting all canonical variable gadgets. Each inversion gadget is connected to the two variable gadgets of the variables involved in the constraint (by copying their values into the two dimensions of the inversion gadget).

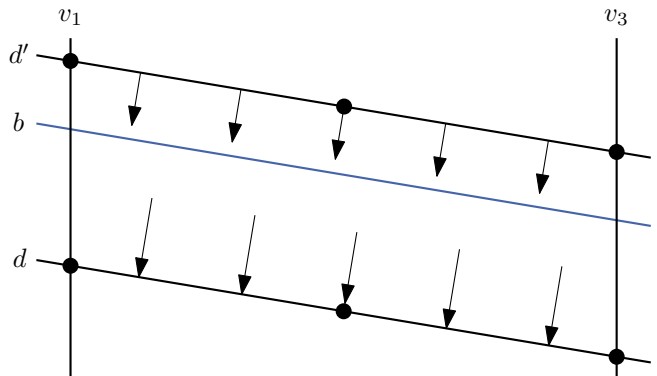

Figure 16: The parallelogram enclosed by the two data lines $d$, $d'$ and the vertical lines $v_1$, $v_3$. The three data points (black) on each data line enforce the gradient in both cells to be orthogonal to the data lines. As a consequence, the breakline $b$ (blue) separating the cells has to be parallel to the data lines.

Finally, we add a cancel gadget for each weak data point such that the cancel gadget contains only this data point but no other data points.

The complete layout can be seen in Figure 17, and an overview over all used gadgets and constructions can be seen in Table 2.

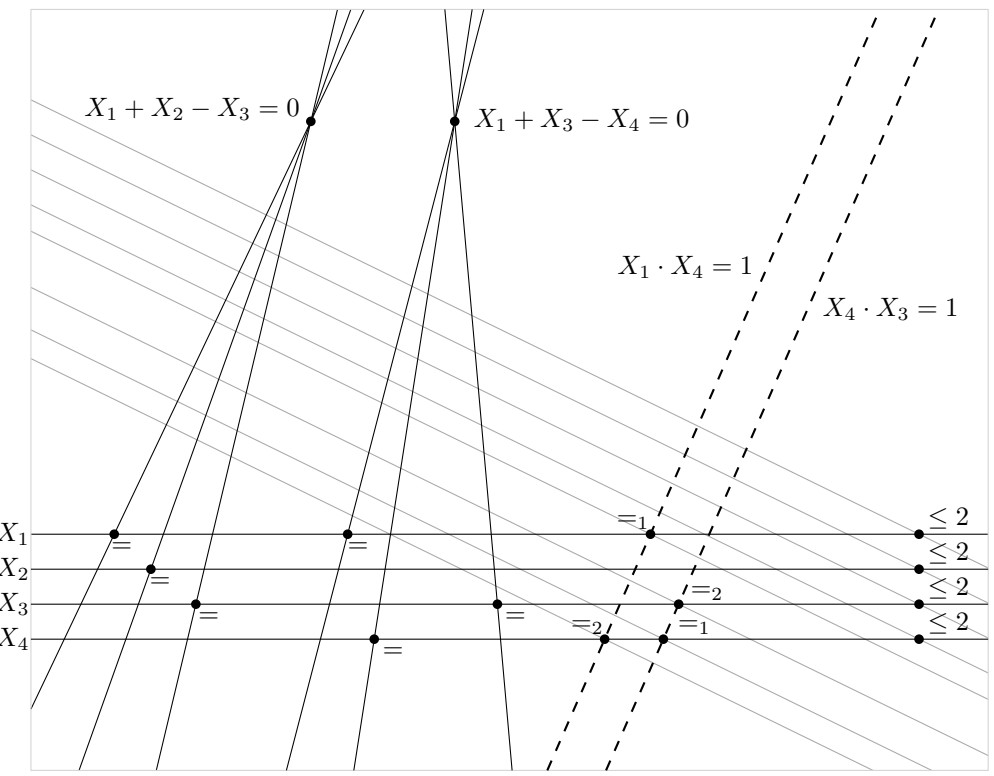

Figure 17: The layout of all gadgets and additional data points for the complete reduction. Each gadget is simplified to a single line for clarity. Solid: Variable gadgets. Dashed: Inversion gadgets. Gray: Cancel Gadgets. A point with label $=_i$ indicates a copy that is only active in output dimension $i$.

*Proof of Theorem 3.* For $\exists\mathbb{R}$-membership we refer to Appendix A. For $\exists\mathbb{R}$-hardness, we reduce from the $\exists\mathbb{R}$-complete problem ETR-INV to TRAIN-F2NN. Given an instance of ETR-INV, we

Table 2: An overview of all parts of the construction.

| Gadget | #Breaklines | #Data Points | Labels |
|---|---|---|---|
| variable gadget | 4 | 37 | $(0,0), (2,2), (3,3), (4,4), (6,6)$ |
| inversion gadget | 5 | 39 | $(0,0), (2,2), (0,3), (3,6), (4,4), (6,6)$ |
| addition | 0 | 1 | $(10,10)$ |
| copy | 0 | 1 | $(-2,6), (6,-2), (6,6)$ |
| cancel gadget | 3 | 24 | $(0,0), (0,-1), (-1,0), (-1,-1)$ |

construct an instance of TRAIN-F2NN with $\gamma = 0$ as described in the previous paragraphs and shown in Figure 17.

Let $m$ be the minimum number of breaklines needed to realize all gadgets of the above construction: We need four breaklines per variable gadget (Lemma 11), five breaklines per inversion gadget (Lemma 15) and three breaklines per cancel gadget (Lemma 16). We obtain the following chain of equivalences, completing our reduction:

> The ETR-INV instance is a yes-instance.
>
> $\Leftrightarrow$ There exists a satisfying assignment of the variables of the ETR-INV instance.
>
> $\Leftrightarrow$ There exists a continuous piecewise linear function fitting all data points of the TRAIN-F2NN instance constructed above and fulfills the conditions of Observation 6 with $m$ breaklines.
>
> $\Leftrightarrow$ There exists a fully connected two-layer neural network with $m$ hidden ReLU neurons fitting all the data points.
>
> $\Leftrightarrow$ The TRAIN-F2NN instance is a yes-instance.

The first and the last equivalence are true by definition.

To see that the second equivalence is true, first assume that there is a satisfying assignment of the variables of the ETR-INV instance. Then these values can be used to find suitable slopes for the variable gadgets, inversion gadgets and cancel gadgets in the construction (recall that the slope is the value plus 1). The superposition of all these gadgets yields the desired continuous piecewise linear function. The function satisfies Observation 6 because, first, the gadgets are built in such a way that functions fitting all data points are constantly zero everywhere except for within the gadgets, and second, the gradient condition is satisfied for each gadget separately and, hence, also for the whole function. For the other direction assume that such a continuous piecewise linear function exists. By Lemmas 16 and 18 the data points enforce exactly the same continuous piecewise linear function as the conceptual data lines and weak data points would. By Lemmas 11, 12 and 15 this continuous piecewise linear function has the shape of the gadgets. Now using that all data points are fit, the slopes of the variable gadgets, and inversion gadgets indeed correspond to a satisfying assignment of ETR-INV.

For the third equivalence, Observation 6 guarantees that such a fully connected two-layer neural network with $m$ hidden ReLU neurons exists. To show the other direction, first note that the function realized by a fully connected two-layer neural network with $m$ hidden ReLU neurons is always a continuous piecewise linear function with at most $m$ breaklines and satisfying the gradient condition. To see that the other condition of Observation 6 is satisfied, note that the only way to fit all data points with a continuous piecewise linear function of this type is such that it is constantly zero outside all the gadgets.

The TRAIN-F2NN instance can be constructed in polynomial time, as the gadgets can be arranged in such a way that all data points (residing on intersections of lines) have coordinates which can be encoded in polynomial length.

The number of hidden neurons $m$ is linear in the number of variables and the number of constraints of the ETR-INV instance. The number of data points can be bounded by $10m$, thus the number of hidden neurons is linear in the number of data points.

As can be gathered from Table 2, the set of used labels is

$$\{(-2,6),(-1,-1),(-1,0),(0,-1),(0,0),(0,3),(2,2), (3,3),(3,6),(4,4),(6,-2),(6,6), (10,10)\}$$

with cardinality 13 as claimed. □

**Remark 23.** Note that if the ETR-INV instance is satisfiable, each variable gadget and inversion gadget in a corresponding solution $\Theta$ to the constructed TRAIN-F2NN instance has a slope of at most 3 in each dimension. Furthermore, no cancel gadget needs to contribute less than $-12$ to satisfy its corresponding weak data point. Thus, there must also be a solution $\Theta'$, where each cancel gadget is symmetric, and thus the function $f(\cdot, \Theta')$ is Lipschitz continuous with a low Lipschitz constant $L$, which in particular does not depend on the given ETR-INV instance. Checking all the different ways how our gadgets intersect, one can verify that $L = 25$ is sufficient.

## C   Algebraic Universality

It remains to prove algebraic universality of TRAIN-F2NN. Intuitively, it suffices to show that the transformations of a solution of an ETR-INV instance to a solution of the corresponding TRAIN-F2NN instance and vice-versa in the above proof of Theorem 3 require only basic field arithmetic, that is, addition, subtraction, multiplication, and division.

**Lemma 24.** *Let $I$ be an instance of* ETR-INV *and $N$ be the instance of* TRAIN-F2NN *built from $I$ by our reduction. We denote by $k$ and $\ell$ the number of variables in a solution of $I$ and $N$ respectively. In* ETR-INV, *these are the variables $X_1, \ldots, X_k$. In* TRAIN-F2NN, *these are the weights and biases of the neural network, in some predefined order. The set $V(I) \subseteq \mathbb{R}^k$ denotes the set of solutions to $I$, and the set $V(N) \subseteq \mathbb{R}^\ell$ denotes the set of solutions to $N$, respectively. Let $\mathbb{F}$ be a field extension of $\mathbb{Q}$. Then,*

$$V(I) \cap \mathbb{F}^k = \emptyset \iff V(N) \cap \mathbb{F}^\ell = \emptyset.$$

*Proof.* We first show that $V(N) \cap \mathbb{F}^\ell \neq \emptyset$ implies $V(I) \cap \mathbb{F}^k \neq \emptyset$: Let $\Theta \in V(N) \cap \mathbb{F}^\ell$ be a solution of the TRAIN-F2NN instance $N$. For each variable $X$ of the encoded ETR-INV instance $I$ there is the canonical variable gadget corresponding to $X$ whose slope $s_X$ satisfies $X = s_X - 1$. There is a unique hidden neuron $v_i$ contributing the first breakline of that variable gadget. Using the notation from Appendix B.1, the slope of this variable gadget is $a_{2,i} \cdot c_{i,1}$, because the variable gadget is horizontal (implying that $a_{1,i} = 0$) and its output is equal in both output dimensions (implying $c_{i,1} = c_{i,2}$). Thus, $X = a_{2,i} \cdot c_{i,1} - 1$, which is clearly in $\mathbb{F}$. Moreover, the vector of all values of the variables carried on the variable gadget (which is a solution as proven in the proof of Theorem 3) is in $V(I) \cap \mathbb{F}^k$, showing that this set is not empty.

We now prove the opposite direction. Let $X_1, \ldots, X_n \in V(I) \cap \mathbb{F}^k$ be a solution of the ETR-INV instance $I$. In our reduction, we place our data points on rational coordinates, and thus all implied data lines can be described by equations with rational coefficients. There exists a unique continuous piecewise linear function $f$ which fits these data points, corresponds to the solution $X_1, \ldots, X_n$, and has the property that all cancel gadgets are symmetric. This function can be realized by a fully connected two-layer neural network. All the gradients of linear pieces in this function can be obtained through elementary operations from the values $X_1, \ldots, X_n$ and rational numbers. Furthermore, all breaklines can be described by equations with coefficients derivable from these same numbers. Thus, there exist weights and biases $\Theta \in \mathbb{F}^\ell$ for the neural network which realize function $f$. As $\Theta$ realizes $f$, it fits all data points, and thus $\Theta \in V(N)$, showing that $\Theta \in V(N) \cap \mathbb{F}^\ell \neq \emptyset$. □

Now Theorem 4, the algebraic universality of TRAIN-F2NN, follows directly from the algebraic universality of ETR-INV (Theorem 8) combined with Lemma 24.