# OpenReview forum: "Training Fully Connected Neural Networks is $\exists\mathbb{R}$-Complete"
_NeurIPS.cc/2023/Conference — NeurIPS 2023 poster_

### Official Review · Reviewer_64NS · 2023-06-22

**Soundness:** 4 excellent
**Presentation:** 4 excellent
**Contribution:** 2 fair
**Rating:** 6
**Confidence:** 4

**Summary:**

The paper investigates the algorithmic complexity of training fully connected ReLU neural networks. It is shown that perfectly fitting two-dimensional data by even a one-hidden layer network with two output neurons is $\exists\mathbb{R}$-complete, namely it is as hard as finding the roots of a multivariate polynomial with integer coefficients.

The paper is well-composed and easy to read, and the findings are interesting even though they are somewhat limited as elaborated in the strengths/weaknesses sections. Overall, my opinion of this paper is positive and I believe it merits acceptance.

**Strengths:**

- Very well-written and easy to follow. It is evident that the authors have put a lot of effort into composing this paper and thoroughly discussing its results, impact and drawbacks.

- The results in this paper are interesting and may suggest that training neural networks is strictly harder (assuming a somewhat acceptable complexity theory hypothesis) than problem in NP. This indicates that certain practices make the worst-case complexity of real-world problems much harder, but the problems are nevertheless solved efficiently. As motivation for future work, it is interesting to study how changing these practices might affect the efficacy in which neural networks are being trained.

**Weaknesses:**

- The paper studies the problem of perfectly fitting the data, while in practice we are interested in getting the training error to become sufficiently small.

- The paper assumes that the learning problem requires classifying at least 13 classes that are two dimensional and are not linearly dependent. While I agree with the authors' claim that the gap between 3 and 13 classes seems immaterial, I still find the two-dimensional output requirement very limiting. This is due to the fact that we can always re-encode the target classes in a different manner which circumvents $\exists\mathbb{R}$-completeness. E.g., we can just encode the classes using the naturals $1,2,\ldots$. The authors point out that a common practice is to use $k$-dimensional standard unit vectors to encode $k$ different classes which may still imply $\exists\mathbb{R}$-completeness, but the study of this is left to future work. To summarize this weakness, the paper cannot currently rule out the possibility that common practices circumvent $\exists\mathbb{R}$-completeness entirely.

**Questions:**

- Line 84: Why do you assume that $\gamma$ is rational and not real?

- Line 109: The theorem statement says that the problem is hard "even" if certain conditions are met. I find this statement very confusing. It sounds like you can replace "There are exactly two" with "There are only two" to make it clearer. Similarly, I would change "The number of hidden neurons is a constant fraction of the number of data points." with "The number of hidden neurons is only a constant fraction of the number of data points.". Lastly, it seems like the last two items are a necessary condition rather than an "even" statement. Specifically, solving for $\gamma=0$ is at least as hard as approximating to some other $\gamma>0$, so it doesn't make sense to use state in line 109 that this holds "even if". Likewise, the assumption of using a ReLU activation is a necessary condition in the analysis from what I understand so this item also does not belong following an "even if" statement. Am I missing something? If so, please clarify the theorem statement.

- Paragraph starting with line 226: It is mentioned that we can add inconsistent data points to modify the value of $\gamma$. Do you mean that this allows us to show a reduction from the $\gamma=0$ case to a general $\gamma>0$ case, and thus the two problems are equivalent? Please clarify.

**Limitations:**

Yes

---

> ### Author Rebuttal · Authors · 2023-08-08
>
> We thank you for carefully reading and assessing our paper and for the valuable feedback.
>
> We agree that our reduction only works for perfect fitting of the data and two-dimensional outputs, as we also discuss in the paper. While the first property is inherent to the nature of the existential theory of the reals, we would like to comment further on one- vs. multi-dimensional outputs. From our perspective, it is very unusual to encode several classes with one-dimensional naturals $1,2,\dots$, since this creates an unnatural and unintended artificial ordering on the classes. We are not aware of this being a common practice. In fact, in practice, one does not only want to distinguish finitely many classes, but also obtain probabilities of data points belonging to the different classes. This naturally calls for one output dimension per class and underlines why people tend to use one-hot encodings in practice. Therefore, multiple output dimensions are practically highly relevant, and we show hardness for this case, even though it remains an open question whether the hardness can be achieved with only one-hot encoded labels.
>
> Answers to the questions:
> 1) Requiring $\gamma$ to be rational is just a technical limitation. As we consider computational complexity in the Turing machine model, a problem instance (including $\gamma$) must be encoded onto the tape of the machine. For rational numbers this is straight-forward, for other real (algebraic) numbers much more involved encodings are necessary: A complication that we want to avoid in favor of simplicity.
>
> 2) Thank you very much for the very good suggestions to improve the wording!
> Regarding the last two items ($\gamma = 0$ and the ReLU activation function), we see that the wording should also be improved. Let us try to clarify: We intend to express that the general decision problem (where an arbitrary $\gamma$ and possibly different activation functions on every neuron are specified as part of the input instance) is ER-hard by showing that a special case ($\gamma = 0$ and ReLU everywhere) is already hard.
> Focusing on the activation function for now, we might also state that the training problem is ER-hard for any class of activation functions that contains ReLU, for example piecewise linear functions. Of course, this does not allow us to easily say something about the hardness of other special cases like some specific piecewise linear function other than ReLU. In the "Activation Function" paragraph in Section 4, we discuss for which types of activation functions we expect a similar argument as ours to work.
>
> 3) The two cases $\gamma=0$ and $\gamma>0$ are not equivalent. Usually, $\gamma>0$ is the harder case, so showing hardness for $\gamma=0$ (as we do in this paper) is stronger. There are in fact settings where there is a difference in complexity, e.g., in the case of a single hidden ReLU neuron, the $\gamma=0$ case is in P [40], but the $\gamma>0$ case is NP-hard [26, 34, 40] (reference numbering as in our original submission). As we mention in the paper, intuitively, a reduction from the $\gamma=0$ case to the $\gamma>0$ case can be achieved by adding inconsistent data points to the instance such that a (a priori known) positive error is unavoidable. To make such a reduction formal, one needs to introduce (mild) additional conditions on the loss function: for example, the loss function needs to satisfy some kind of monotinicity and preserve small encoding sizes in the bit model. Despite being mild and satisfied by most loss functions that are computable on the real RAM, these conditions are somewhat technical, which is why we decided not to go into detail here in the paper.

---

> > ### Comment · Reviewer_64NS · 2023-08-18
> > **Post-rebuttal response**
> >
> > Dear authors,
> >
> > Thank you very much for your detailed answer to each of my questions. I find your answers satisfying and wish you good luck in the future.

---

### Official Review · Reviewer_LVwc · 2023-07-04

**Soundness:** 3 good
**Presentation:** 4 excellent
**Contribution:** 4 excellent
**Rating:** 7
**Confidence:** 3

**Summary:**

The paper proves ER-completeness of training (empirical risk minimization) a fully connected two-layer ReLU network with two inputs and two outputs. The result contributes to the understanding of theoretical complexity of neural network training at an important, fundamental level. The authors also did a good job explaining the intricacies of the achieved type of hardness result with respect to implications and applicability in the broader neural network training context. The true main, and (unavoidably) very technical part of the paper, i.e., the actual proofs, are delegated to a "supplementary material" document that is twice as long as the "main" paper.

-- update: I have read and acknowledged all other reviews and the authors rebuttals, see discussion. --

**Strengths:**

The result of this paper constitutes and important and fundamental contribution to the understanding of the complexity of training neural networks (to global optimality). Besides the main result itself, the paper's strength lie in its clarity of presentation and the discussion of context, limitations and implications.

**Weaknesses:**

It appears to have become common practice to submit papers to NeurIPS (and ICML) whose actual, main content is put in a separate "Supplementary Material" document whose length far exceeds that of the supposed main paper. Unfortunately, this paper is no exception. I consider this a weakness because this format bears the danger of the formally most important parts of the work not being reviewed thoroughly due to the short review period and high review load of reviewers at these conferences. I cannot exclude myself from this -- I simply did not have the time to rigorously check all the details in the long supplementary document, and therefore cannot give a definitive answer regarding the proofs' correctness beyond "believing" everything appears to be in order. In this regard, I cannot help but wonder if a full journal paper would not be the better way to publish results that simply do not fit into the 9-page limit...

**Questions:**

It seems that, while generally helpful, the proof outline in the "main" paper does not quite bring across why the reductions works. It is well-written and enables one to grasp the general concepts (gadgets etc) used for the reduction, but perhaps the authors could add some clarification with respect to the "crucial" step, i.e., why their encoding can be shown to answer the ETR instance if and only if the Train-F2NN problem can be fitted with 0 loss?

**Limitations:**

The authors adequately commented on applicability/meaning of their result and, thus, its inherent limitations.

---

> ### Author Rebuttal · Authors · 2023-08-08
>
> We thank you for carefully reading and assessing our paper and for the valuable feedback.
>
> As other reviewers had similar suggestions, we plan to enhance the proof ideas section, especially concerning the interplay of the gadgets as well as the equivalence of the training problem and the ETR-INV instance.
>
> To answer your concrete question, in our reduction, the slopes of some linear pieces of the neural network correspond to (real-valued) variable assignments of the ETR-INV instance. The gadgets ensure that all the constraints of the ETR-INV instance are satisfied. Hence, there exists a satisfying assignment for the ETR-INV instance if and only if there exists a piecewise linear function with the correct slopes, which in turn happens if and only if the training problem admits a zero-error solution. This is the concluding step of the reduction.
>
> In principle, we share your concerns about the publication culture and review process for complicated theoretical results at conferences like NeurIPS. However, these conferences are the venues with the highest visibility in the ML community and therefore submitting and publishing there is the best way to disseminate and advertize important results in a timely manner. From our perspective, this also applies to important theoretical results, even if they require long appendices. Of course, since there is no way of incorporating all the technical details into the main part, we gave our best to provide context and intuitions in the main paper and will, as promised above, improve this further using the constructive feedback from the reviews.

---

> > ### Comment · Reviewer_LVwc · 2023-08-10
> >
> > I thanks the authors for responding to concerns raised by the other reviewers and myself. To me, all have been adequately addressed, and given that the paper was already well-written, I trust that the suggested clarity improvements will be incorporated satisfactorily into the final version, as promised by the authors in their replies. Thus, I uphold my vote and recommend accepting this paper.

---

### Official Review · Reviewer_QSnu · 2023-07-05

**Soundness:** 3 good
**Presentation:** 3 good
**Contribution:** 3 good
**Rating:** 6
**Confidence:** 4

**Summary:**

This paper shows that training a neural network with two input neurons and two output neurons is ER complete where ER stands for the existential theory of the reals.

**Strengths:**

A new idea (reduction) to prove hardness result. The paper is generally well written and provides a good cover of the (dense) related work.

**Weaknesses:**

In the end, the results in the paper point more to the limitation of current computational hardness results than to the limitation of deep learning. The task of finding the exact minimizer and the inability to prove hardness results for the approximate case diminish the relevance of the results to modern ML. Personally, I'm not a fan of ER but I think the reductions could find uses elsewhere.
The paper is somewhere between "accept" to "weak accept" and I'm willing to raise the score if the authors provide convincing responses.

There are several issues with the writing such as:
1) Definition 2 is very cumbersome. Consider shortening it as it done in the Blum Rivest paper.
2) "However, when using gradient descent
 we usually do not get any guarantees on the quality of the obtained solutions or on the time it takes to
 compute them [30]". This is misleading as there are many (proven) results about GD finding the global minimum. [30] is one example one GD does not work well, but there are many other results for which it does work well such as "Globally optimal gradient descent for a convnet with gaussian inputs".
3) "Thus it would be very desirable to use other methods, like SAT- or mixed-integer
programming solvers that are reliable and widely used in practice for many NP-complete problems."
What does "reliably" mean here? I'm sure that there are bad examples for these heuristics just as there are for GD.
4) The elephant in the room which the authors largely ignore is the huge success of gradient-based optimization in training neural networks with tens of millions of neurons.
5) Definition 1 is strange. Consider using the standard matrix notation for neural networks.
6) There seems to be a redundancy between the average case analysis paragraph and the "Let us stress" paragraph.
7) I would put still more effort in explaining the ideas behind the main reduction and exemplifying it with a small example.

**Questions:**

What happens to the hardness result if we only consider neural networks with limited precision? Say any weight or bias is represented by at most 10 bits?

**Limitations:**

I think one of the biggest issues is not addressing the success of gradient-based optimization in the paper.
The authors might want to talk more about improper learning which are typically used and make the learning task easier.

---

> ### Author Rebuttal · Authors · 2023-08-08
>
> We thank you for carefully reading and assessing our paper and for the valuable feedback.
> We also thank you very much for the detailed list of suggestions to improve the presentation of our paper, which we happily take into account. See below for some detailed responses:
>
> 1+5: We agree and plan to simplify the definitions as we do not need the full generality in our fully connected two-layer setting. (This was also suggested by another reviewer.)
>
> 2: This is a good point and we will refine our discussion here to also include the positive examples. In fact, it is a very interesting research question how/whether these proven positive examples for GD can be used for other, more geometric ER-complete problems.
>
> 3: What we mean by "reliably" is that they always terminate with an optimal solution in finite (but sometimes exponential) time. They are not just heuristics, in contrast to GD (in the general case, acknowledging the positive examples mentioned by the reviewer in the previous question). So, for SAT-solving or MIP, there are no "bad examples" in terms of solution quality. There are just "bad examples" in terms of running time, but for many problems also these examples are rare. We agree that "reliable" is not a good term here and will revise this discussion in the final version.
>
> 4: You are right that this deserves a more prominent discussion. We plan to extend the discussion in Section 4 (Implications of ER-Completeness) to make this more explicit and to stress that our results are of a purely theoretical nature. As we say, neither does ER-hardness rule out successful heuristics nor does ER-membership directly lead to good off-the-shelf solution strategies. We are still convinced that a thorough understanding of the theoretical worst-case complexity of the training problem is a highly important contribution to the field. After all, a good theoretical understanding of the "exact training" case may also inspire future work to better understand "approximate training".
>
> 6: Good find. We fix this when incorporating your suggestions regarding (4).
>
> 7: Enhancing the proof ideas section was suggested by other reviewers as well, especially concerning the interplay of the gadgets and the equivalence of the training problem and the ETR-INV instance.
>
> Some thoughts regarding your question about limited precision:
> This is a very good question and we discuss it a little bit in Section 4 ("Required Precision of Computation"). From a purely theoretical point of view, we suspect that limiting each weight and bias by 10 bits would lead to NP-membership, as these bits could be "guessed" by a nondeterministic Turing machine. But we did not investigate this thoroughly and have no formal proof of this claim.
> From a less (but still) theoretical perspective, potential membership in NP of the approximate version could spark hope that off-the-shelf tools like SAT- or ILP-solvers would be applicable. A naive approach (and we do not know of anything better) requires one Boolean variable per bit for each weight/bias. This does not seem viable, especially compared to the widely used and successful GD-based approaches.
> To conclude: This restriction would break our ER-hardness reduction (ER-hardness and high precision often go hand-in-hand) but also not immediately lead to new approaches for training in practice.

---

> > ### Comment · Reviewer_QSnu · 2023-08-13
> > **Repsonse to authors**
> >
> > I keep my score and support acceptance. I encourage the authors to elaborate much more on the proof ideas.

---

### Official Review · Reviewer_2pEa · 2023-07-06

**Soundness:** 3 good
**Presentation:** 2 fair
**Contribution:** 3 good
**Rating:** 5
**Confidence:** 2

**Summary:**

This paper strengthens a result by Abrahamsen, Kleist and Miltzow [Neurips 21] by showing that empirical minimization problem of two-layer neural network is \exist R-hard. They further show that arbitrary algebraic number is required as optimal weights even with rational data points. The example include a ReLU neural network with two inputs and two outputs.


**Strengths:**

- The paper is well-written and the related works have been discussed in great details.
- The reduction of training two-layer neural network to ETR-INV is novel.
- The construction of inversion gadget in appendix B.3.4 is quite interesting to learn.

**Weaknesses:**

- Definition 1 which defines neural networks from a directed acyclic graph seems to be redundant as the paper is devoted to analyze a two-layer fully connected neural networks, whose structure is much simpler.
- The structure of the paper can be improved. The majority of the papers is to introduce definitions and discussing related works, while the main proof idea only starts at page 7. It would be better if the authors can elaborate on the proof idea. From my perspective, the main theoretical contribution in this paper is the reduction of training two-layer neural network to ETR-INV, while the details how they are achieved are largely left in the appendix.

**Questions:**

See weakness.

**Limitations:**

Yes.

---

> ### Author Rebuttal · Authors · 2023-08-08
>
> We thank you for carefully reading and assessing our paper and for the valuable feedback.
>
> Our definition of neural networks as directed acyclic graphs is in line with the definition by Abrahamsen, Kleist and Miltzow (NeurIPS 21) whose result we strengthen. While this generality makes sense in their case (their reduction crucially depends on which "edges" are present), we agree that the definition in our fully connected setting can be simplified significantly and we plan to do so in the revision.
>
> We consent that our main contribution is the reduction from ETR-INV to training two-layer neural networks, thereby proving ER-hardness. Regarding the structure of the main body, we plan to enhance the proof ideas section, especially concerning the interplay of the gadgets as well as the equivalence of the training problem and the ETR-INV instance (this was also suggested by other reviewers).
>
> Unfortunately, we see no way to incorporate all details of the reduction into the main body without sacrificing clarity. In our opinion, a thorough review of the related work and a detailed discussion of the strengths and weaknesses should be placed prominently in the paper as they are probably more interesting to a wider audience than the details of the proof.

---

> > ### Comment · Reviewer_2pEa · 2023-08-14
> > **Response to authors**
> >
> > Thanks for the response. I would like to keep the score.

---

### Decision · Program_Chairs · 2023-09-21

**Decision:**

Accept (poster)

**Comment:**

This paper proves ER-hardness results for optimally training two-layer neural network architectures. Although there are some limitations to this type of worst-case analysis of machine learning problems, these limitations are clearly discussed in the paper. More generally, the reviewers generally found this work, including its relation to previous work, to be reasonably well-explained and the technical contribution of this work to be novel. Based on this, I recommend acceptance.